# A parabrachial to hypothalamic pathway mediates defensive behavior

**Fan Wang**[1,2†], **Yuge Chen**[1,2†], **Yuxin Lin**[1,2‡], **Xuze Wang**[1,2‡], **Kaiyuan Li**[1,2], **Yong Han**[1,2], **Jintao Wu**[1,2], **Xingyi Shi**[1,2], **Zhenggang Zhu**[1,2], **Chaoying Long**[1,2], **Xiaojun Hu**[1,3], **Shumin Duan**[1,2,3,4,5,6]*, **Zhihua Gao**[1,2,3]*

[1]Department of Neurobiology and Department of Neurology of Second Affiliated Hospital, Zhejiang University School of Medicine, Hangzhou, China; [2]Liangzhu Laboratory, Zhejiang University Medical Center, MOE Frontier Science Center for Brain Science and Brain-machine Integration, State Key Labotatory of Brain-machine intelligence, Zhejiang University, Hangzhou, China; [3]NHC and CAMS Key Laboratory of Medical Neurobiology, Zhejiang University, Hangzhou, China; [4]Institute for Translational Brain Research, MOE Frontiers Center for Brain Science, Fudan University, Shanghai, China; [5]The Institute of Brain and Cognitive Sciences, Zhejiang University City College, Hangzhou, China; [6]Chuanqi Research and Development Center of Zhejiang University, Hangzhou, China

*For correspondence:
duanshumin@zju.edu.cn (SD);
zhihuagao@zju.edu.cn (ZG)

†These authors contributed equally to this work
‡These authors also contributed equally to this work

Competing interest: The authors declare that no competing interests exist.

**Abstract** Defensive behaviors are critical for animal's survival. Both the paraventricular nucleus of the hypothalamus (PVN) and the parabrachial nucleus (PBN) have been shown to be involved in defensive behaviors. However, whether there are direct connections between them to mediate defensive behaviors remains unclear. Here, by retrograde and anterograde tracing, we uncover that cholecystokinin (CCK)-expressing neurons in the lateral PBN (LPB[CCK]) directly project to the PVN. By in vivo fiber photometry recording, we find that LPB[CCK] neurons actively respond to various threat stimuli. Selective photoactivation of LPB[CCK] neurons promotes aversion and defensive behaviors. Conversely, photoinhibition of LPB[CCK] neurons attenuates rat or looming stimuli-induced flight responses. Optogenetic activation of LPB[CCK] axon terminals within the PVN or PVN glutamatergic neurons promotes defensive behaviors. Whereas chemogenetic and pharmacological inhibition of local PVN neurons prevent LPB[CCK]-PVN pathway activation-driven flight responses. These data suggest that LPB[CCK] neurons recruit downstream PVN neurons to actively engage in flight responses. Our study identifies a previously unrecognized role for the LPB[CCK]-PVN pathway in controlling defensive behaviors.

## Editor's evaluation

In this study, the authors revealed that activation of LPB CCK-expressing neurons could drive flight-to-nest behavior and increase sympathetic output while inhibiting the cells reduces the behavioral response to predator exposure and visual predatory cues. They further found that LPB CCK neurons project to the PVN area, and activation of this pathway caused similar behavioral changes. Lastly, activating PVN glutamatergic cells can also induce flight. The evidence is solid, the study is valuable, and it will be of interest to the affective and circuit neuroscience fields as it provides insights into a novel parabrachial to hypothalamus pathway that could potentially mediate threat avoidance behavior.

## Introduction

Living in an environment full of diverse threats, animals develop a variety of defensive behaviors for survival during evolution (*Anderson and Adolphs, 2014*; *Blanchard et al., 2001*; *Crawford and Masterson, 1982*; *Fanselow and Bolles, 1979*; *LeDoux, 2012*; *Yilmaz and Meister, 2013*). From

the perspective of predator and prey interactions, defensive responses are generally divided into two categories: to avoid being discovered (such as freezing or hiding behaviors) or being caught (such as escaping or attacking behaviors) (*Crawford and Masterson, 1982*; *Fanselow and Bolles, 1979*; *Yilmaz and Meister, 2013*). Which type of defensive behavior is selected depends on many factors, including the intensity and proximity of the threat, as well as the surrounding context. In general, escape is triggered when animals are faced with an imminent threat with a shelter nearby (*Eilam, 2005*; *Lin et al., 2023*; *Perusini and Fanselow, 2015*; *Sun et al., 2020b*; *Zhang et al., 2018*).

Several brain regions, including the periaqueductal gray (PAG), amygdala, and medial hypothalamic zone (MHZ), have been implicated in mediating escape behaviors (*Evans et al., 2018*; *Shang et al., 2018*; *Shang et al., 2015*; *Silva et al., 2013*; *Tovote et al., 2016*; *Wang et al., 2015*; *Wang et al., 2021b*; *Wei et al., 2015*). Recent studies have highlighted a role for the hypothalamus, in particular the MHZ, including the anterior hypothalamic nucleus (AHN), the dorsomedial part of ventromedial hypothalamic nucleus (VMHdm) and the dorsal pre-mammillary nucleus (PMd), in mediating escape behaviors. For example, a subset of VMH neurons collaterally project to the AHN and PAG to promote both escape and avoidance behaviors (*Wang et al., 2015*). Moreover, PMd neurons project to the dorsolateral periaqueductal gray region (dlPAG) and the anteromedial ventral thalamic region (AMV) also control escape behaviors (*Wang et al., 2021b*). In addition, PVN, an area enriched with endocrine neurons, has also been associated with escape behaviors (*Mangieri et al., 2019*). Optogenetic activation of Sim1[+] neurons in the PVN triggers escape and projections from the PVN to both the ventral midbrain region (vMB) and the ventral lateral septum (LSv) mediate defensive-like behaviors, including hiding, escape jumping and hyperlocomotion (*Mangieri et al., 2019*; *Xu et al., 2019*). Despite a growing number of studies illustrating the importance of PVN in mediating defensive behaviors, upstream inputs that transmit the danger signals to the PVN remain unclear (*Isosaka et al., 2015*; *Penzo et al., 2015*).

Located in the dorsolateral pons, PBN, serves as an important relay station for sensory transmission (*Fulwiler and Saper, 1984*). While PBN is best known for various sensory processes to protect the body from noxious stimuli, emerging evidence suggests that it may act as a key node in mediating defensive behaviors (*Campos et al., 2018*; *Day et al., 2004*; *Han et al., 2015a*). Electrical lesions of the PBN or local microinjections of kainic acid into the PBNinduced defensive behaviors (*Mileikovskii and Verevkina, 1991*). In addition, exposure of the olfactory predator cue, trimethylthiazoline (TMT), increased c-fos expression in the LPB (*Day et al., 2004*). More importantly, calcitonin gene-related peptide (CGRP) neurons, located in the external lateral subnuclei of PBN (PBel), have been shown to mediate alarm responses and defensive behaviors under stressful or threatening circumstances via projections to the amygdala and the bed nucleus of stria terminalis (BNST) ( *Han et al., 2015a*; *Zhang et al., 2020*). While both PBN and PVN neurons appear to be involved in defensive responses, it remains unclear whether they connect with each other to mediate defensive behaviors (*Fulwiler and Saper, 1984*).

Both CCK and CCK receptor-expressing neurons have been implicated in defensive behaviors (*Bertoglio et al., 2007*; *Chen et al., 2022*; *Wang et al., 2021a*). PVN neurons also express CCK receptors and are involved in defensive behaviors (*Mangieri et al., 2019*; *O'Shea and Gundlach, 1993*; *Xu et al., 2019*), raising a possibility that upstream CCK inputs to the PVN (*Meister et al., 1994*) may be implicated in defensive behaviors. In an attempt to test this possibility, we injected the Cre-dependent retrograde tracer (AAV2-Retro-DIO-EYFP) into the PVN of *Cck-cre* mice and found that the LPB is an important upstream CCKergic input to the PVN. We further showed that LPB[CCK] neurons were recruited upon exposure to various threat stimuli. Optogenetic activation of either LPB[CCK] neuronal somas or their projections to the PVN promotes defensive responses, whereas their inhibition attenuates defensive-like behaviors, suggesting an essential and sufficient role for LPB[CCK]-PVN pathway in mediating defensive behaviors.

## Results
### LPB[CCK] neurons provide monosynaptic glutamatergic projections to the PVN

To identify upstream CCKergic inputs to the PVN and visualize the range of viral infection, we injected a mixture of retrograde viral tracer (AAV2-Retro-DIO-EYFP, hereafter referred to as AAV2-Retro) and

CTB 555 into the PVN of knock-in mice expressing Cre recombinase at the *CCK* Locus (*Cck-ires-cre*, referred to as *Cck-cre* hereafter; *Figure 1A*). We observed abundant EYFP$^+$ neurons in the LPB, medial orbital cortex (MO), and cingulate cortex (CC), with scattered EYFP$^+$ cells in the PAG and dysgranular insular cortex (DI) (*Figure 1B–H*). Since LPB is a region critical for sensory signal processing, we mainly focused on the LPB in the following study (*Fulwiler and Saper, 1984*; *Garfield et al., 2014*).

LPB contains several subregions with diverse peptide-expressing neuronal subtypes and CCK$^+$ neurons have been shown to primarily locate within the superior lateral PB (PBsl; *Garfield et al., 2014*). To dissect the identity of CCK$^+$ neurons in the LPB, we performed in situ hybridization in *Cck-cre::Ai14* mice using probes for *Slc17a6* and *Slc32a1*, markers specific to glutamatergic and GABAergic neurons (*Figure 1—figure supplement 1A–C*). We found that LPB$^{CCK}$ neurons (84.0% ± 1.7%) predominantly expressed *Slc17a6*, with a tiny subpopulation (2.0% ± 1.2%) expressing *Slc32a1*, suggesting that LPB$^{CCK}$ neurons were mostly glutamatergic neurons (*Figure 1—figure supplement 1D–E*). In addition, co-labeling with another neuropeptide, CGRP, revealed minimal co-localization (2.2% ± 0.37%) between CCK and CGRP (*Figure 1—figure supplement 1F–G*), suggesting primary separation between these two subpopulations of neurons. Collectively, these data demonstrate that LPB$^{CCK}$ neurons comprise predominantly glutamatergic neurons, which barely overlap with CGRP neurons.

To verify the anatomical connections between the LPB and PVN, we injected the anterograde viral tracer (AAV-hSyn-FLEx-mGFP-2A-Synaptophysin-mRuby) into the LPB of *Cck-cre* mice (*Figure 1I–J*). We observed prominent GFP$^+$ fibers and mRuby$^+$ bouton-like structures, reminiscent of axonal terminals, within the PVN (*Figure 1K*) and other brain regions (*Figure 1—figure supplement 2A–L*). Next, we injected the Cre-dependent adeno-associated virus (AAV) expressing channelrhodopsin-2 virus (AAV2/9-DIO-ChR2-EYFP) into the LPB of *Cck-cre* mice (*Figure 1L*) and optogenetically stimulated the axonal terminals of virus-labeled LPB$^{CCK}$ neurons within the PVN in brain slices. We successfully recorded light stimulation-induced excitatory postsynaptic currents (EPSCs), rather than inhibitory postsynaptic currents (IPSCs). These EPSCs were blocked in the presence of the sodium channel antagonist tetrodotoxin (TTX), but rescued by the potassium channel antagonist 4-Aminopyridine (4-AP) (*Figure 1M–N*). Further, NMDA receptor antagonist AP5, and AMPA receptor antagonist CNQX also blocked light stimulation-induced EPSCs (*Figure 1O–P*), validating that PVN neurons receive monosynaptic excitatory innervations from LPB$^{CCK}$ neurons.

## Photostimulation of LPB$^{CCK}$ neurons induces aversion and defensive behaviors

LPB$^{CCK}$ neurons have been shown to regulate glucose homeostasis and body temperature (*Garfield et al., 2014*; *Yang et al., 2020*). However, it remains unclear whether activation of LPB$^{CCK}$ neurons may elicit direct behavioral changes. Before optic stimulation in vivo, we first tested the efficacy of optogenetic stimulation by whole-cell recording in ChR2-expressing LPB$^{CCK}$ neurons. We observed that 5–20 Hz blue laser pulses induced time-locked action potential firing, with 20 Hz inducing maximal firing capacities. We then chose 20 Hz for the following in vivo stimulation (*Figure 2A*).

We injected the control or ChR2-expressing viruses (AAV2/9-EF1a-DIO-EYFP or AAV2/9-EF1a-DIO-ChR2-EYFP) into the LPB of *Cck-cre* mice, followed by optical fiber implantation and optic stimulation (*Figure 2B–D*). Since LPB has been shown to mediate aversion (*Chiang et al., 2019*), we first tested whether optogenetic activation of LPB$^{CCK}$ neurons affects aversion using the real-time place aversion (RTPA) test (*Figure 2E*). We found that activation of LPB$^{CCK}$ neurons significantly reduced the duration of mice in the laser-paired chamber, accompanied by rapid running or flight to the other laser-unpaired chamber, indicating an obvious aversion-like avoidance and/or fear-related defensive-like behaviors (*Figure 2F–G*).

To further investigate whether LPB$^{CCK}$ neurons plays a role in defensive behaviors, we used a well-established flight-to-nest behavioral test by putting a nest in the corner of an arena to test the ability of mice to actively search for hiding (*Figure 2H*). Of note, activating LPB$^{CCK}$ neurons induced robust flight-to-nest behavior in mice, with much shorter latency and faster speed running towards the nest (latency: EYFP, 87.71 ± 22.38 s vs. ChR2, 7.571 ± 0.8123 s; speed: EYFP, 184.4% ± 30.4% vs. ChR2, 361.7% ± 31.5%), followed by longer stay in the nest (EYFP, 63.906% ± 3.645% vs. ChR2, 97.26% ± 2.737%; *Figure 2I–K*). These data suggest that activation of LPB$^{CCK}$ neurons induces aversion-like avoidance and defensive-like flight-to-nest behaviors.

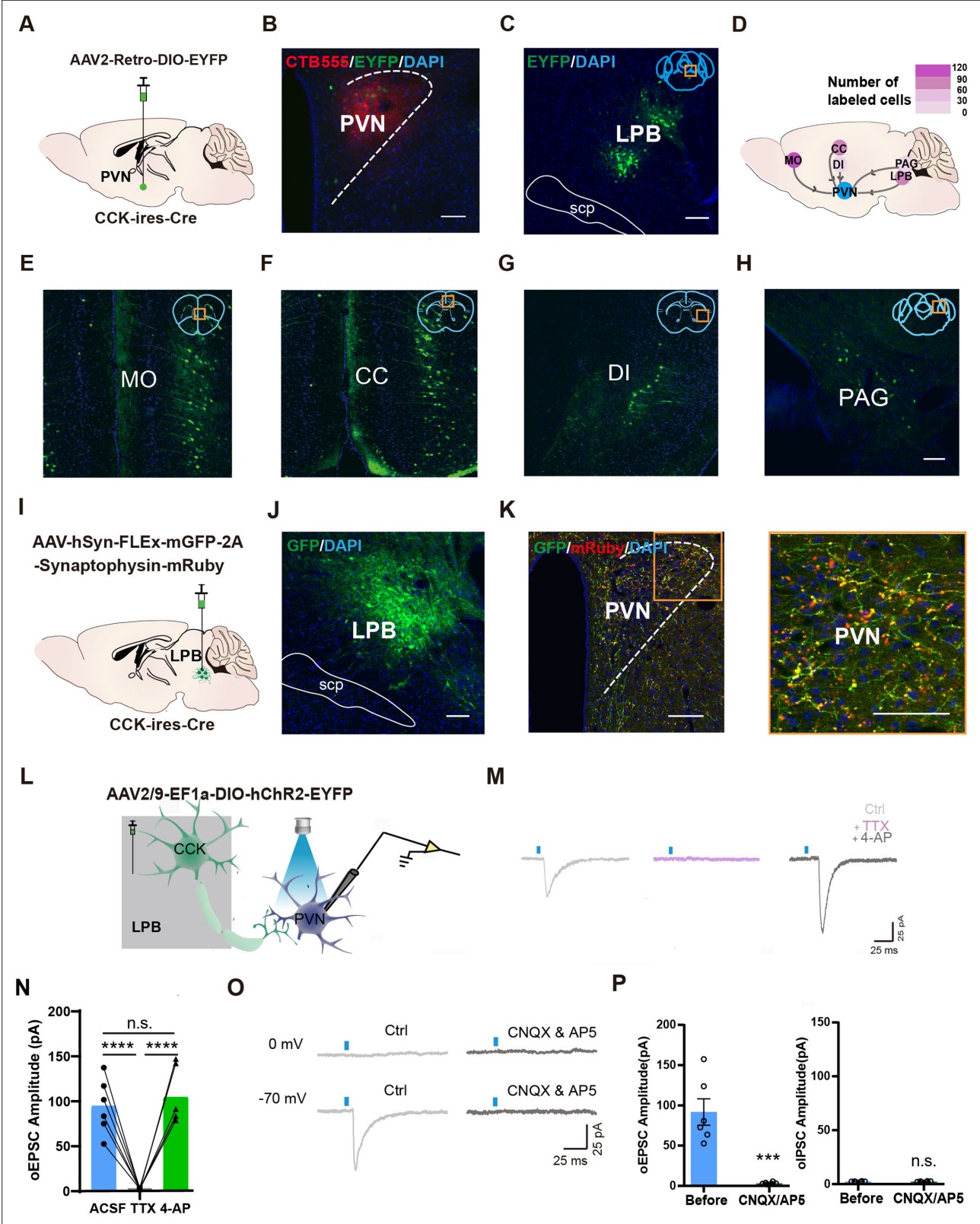

**Figure 1.** LPB^CCK neurons project to the PVN. (**A**) Scheme of viral strategy for retrograde tracing from the PVN in *Cck-cre* mice using AAV2-Retro virus. (**B**) Representative image showing the injection site as marked by CTB555. (**C**) Representative histological images of EYFP⁺ neurons in the LPB. (**D–H**) A heatmap (**D**) demonstrating the distribution of EYFP⁺ neurons in the MO (**E**), CC (**F**), DI (**G**), PAG (**H**), n = 3 mice. (**I–J**) Anterograde tracing of LPB^CCK neurons using AAV-hSyn-FLEX-mGFP-2A-Synaptophysin-mRuby virus. (**K**) Projections of LPB^CCK neurons to the PVN. The right panel shows a magnified

*Figure 1 continued on next page*

*Figure 1 continued*

view of boxed area; scale bar, 100 μm; n = 3 mice. (**L**) Schematic of recording from PVN cells after optogenetic activation of LPB^CCK axonal terminals. (**M**) Representative traces of light-evoked EPSCs recorded from PVN neurons following light stimulation of LPB^CCK axonal terminals in the presence of ACSF (Ctrl), TTX (1 μM) and 4-AP (100 μM). (**N**) Quantification of excitatory postsynaptic currents (EPSCs) from identified PVN neurons receiving inputs from the LPB^CCK neurons in the presence of ACSF (Ctrl), TTX (1 μM) and 4-AP (100 μM). (ACSF vs. TTX ****p < 0.0001; TTX vs. 4-AP ****p < 0.0001; ACSF vs. 4-AP p > 0.9999, one-way ANOVA Bonferroni's multiple comparisons test). (**O**) Representative traces of light-evoked EPSCs recorded from PVN neurons following light stimulation of LPB^CCK axonal terminals in the presence of CNQX (20 μM) and AP5 (50 μM) (n = 9 neurons). (**P**) Quantification of EPSCs and inhibitory postsynaptic currents (IPSCs) from identified PVN neurons receiving inputs from the LPB^CCK neurons. (oEPSC, p = 0.0003 t = 5.378 df = 10; oIPSC, p = 0.8793 t = 0.1558 df = 10; unpaired *t* test).

The online version of this article includes the following source data and figure supplement(s) for figure 1:

**Source data 1.** Quantification of the labeled cells, EPSCs and IPSCs.

**Figure supplement 1.** Distribution and identification of CCK neurons in the LPB.

**Figure supplement 1—source data 1.** Quantification of LPB CCK-tdT cells.

**Figure supplement 2.** Anterograde mapping of LPB^CCK neurons.

---

Defensive behaviors are usually accompanied by changes in the autonomic nervous system, such as increased heart rates, dilated pupil size and elevated cortisol levels (***Dong et al., 2019***; ***Gross and Canteras, 2012***; ***Wang et al., 2015***). We found that photostimulation of LPB^CCK neurons also significantly increased heart rates and enlarged pupil diameters, along with elevated plasma levels of corticosterone (***Figure 2L–P***), suggesting that activating LPB^CCK neurons induces fast sympathetic changes, leading to an arousal state accompanying the defensive responses in mice (***Salay et al., 2018***).

Since activation of LPB^CCK neurons induced fear-associated flight-to-nest behaviors and long-term fear may trigger anxiety-like states in animals (***Tovote et al., 2005***; ***Yang et al., 2016***), we also examined whether prolonged activation of LPB^CCK neurons induces anxiety-like behaviors (***Figure 2—figure supplement 1A***). After 10 min of light stimulation, mice injected with ChR2 exhibited significantly reduced center time and center entries in the open field tests without affecting total distance, and decreased open arm entries and time in the elevated plus maze tests (***Figure 2—figure supplement 1B–J***), suggesting that prolonged activation of LPB^CCK neurons also induces anxiety-like behaviors in mice.

## LPB^CCK neurons encode threat stimuli-evoked flight behaviors

LPB neurons were shown to be activated when animals are in proximity to dangerous stimuli (***Campos et al., 2018***; ***Day et al., 2004***; ***Han et al., 2015b***). To track the endogenous activities of LPB^CCK neurons in vivo, we injected the Cre-dependent fluorescent calcium indicator AAV-DIO-GCaMP7s into the LPB of *Cck-cre* mice, and recorded the calcium responses by fiber photometry (***Figure 3A–C***). Consistent with previous observation (***Yang et al., 2020***), we observed elevated calcium responses of LPB^CCK neurons by heat stimuli (43 °C) (***Figure 3—figure supplement 1A–D***). We then tested the response of these neurons in a predator-exposure assay, in which an awake but restrained rat was placed at one end of a rectangular arena (***Reis et al., 2021***; ***Weisheng et al., 2021***), and a mouse was placed at the other end, away from the rat. After perceiving the presence of the rat, mice usually exhibit risk assessment behaviors by curiously approaching and investigating the rat (***Olivier et al., 1991***). We observed that the calcium signals of LPB^CCK neurons gradually rise when mice approached the rat. At the moment of escape initiation, when mice turned back and dashed away from the rat, the calcium signals reached a peak (***Figure 3D–G and L***). However, the signals ramped down as mice gained distance from the rat. Notably, activities of LPB^CCK neurons were unaffected in the presence of a toy rat (***Figure 3—figure supplement 1E–H***).

In the presence of the visual predatory cue, such as appearing and expanding looming shadows to mimic an approaching predator (***Yilmaz and Meister, 2013***), LPB^CCK neurons also showed increased calcium signals (***Figure 3H***). Similar to rat exposure assay, we also tested the response of these neurons in the looming assay, the elevated calcium signals reached peak when the animal initiated escape, but reduced once it ran into the nest (***Figure 3H–K and M***). LPB^CCK neurons also showed elevated calcium signals when mice were exposed to the olfactory predatory cue, TMT odor (***Figure 3—figure supplement 1–L***). Correlative analysis revealed that the elevation of the calcium signal was correlated with the onset of escape responses (***Figure 3N–O***). Thus, threatening stimuli of different sensory

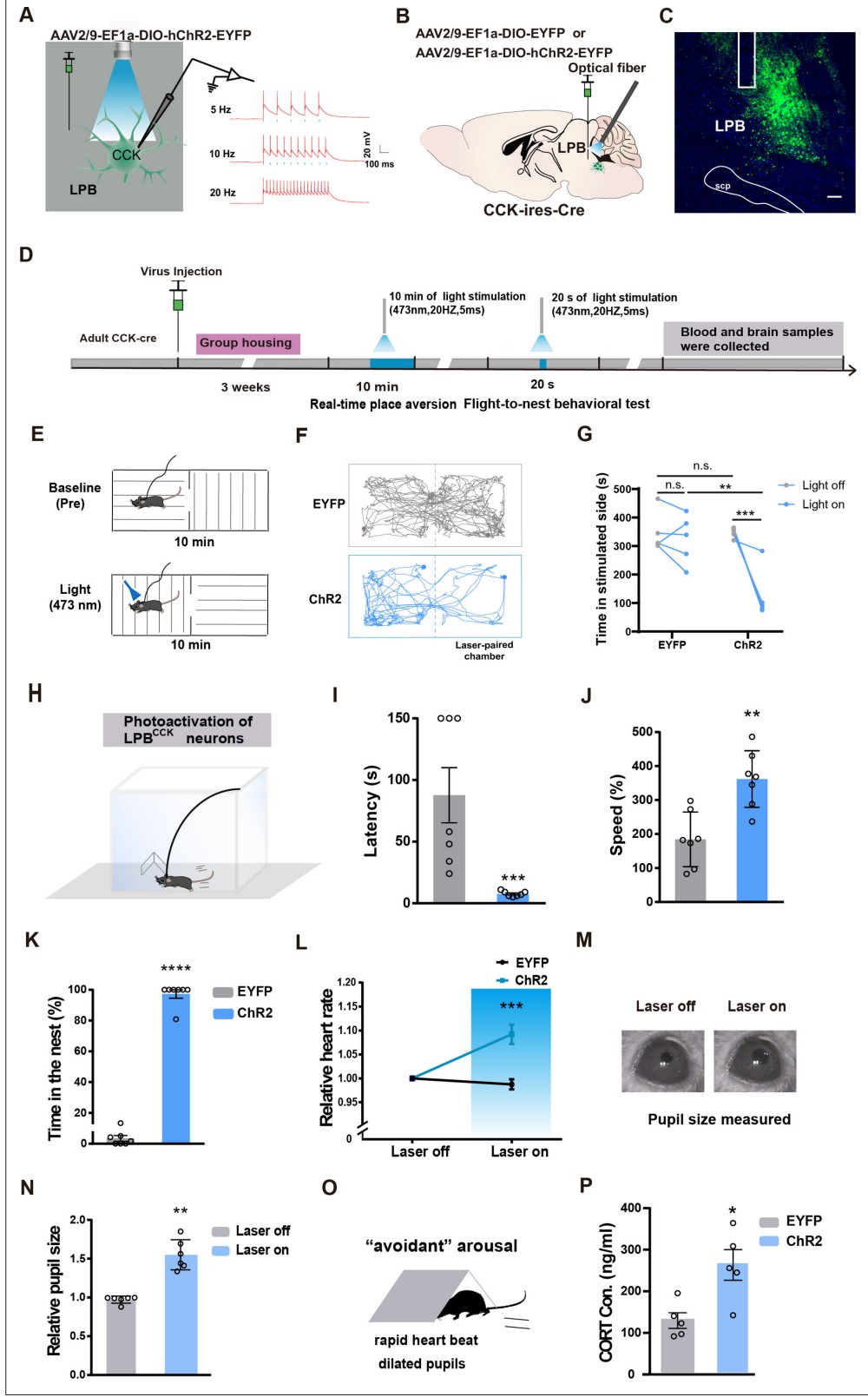

**Figure 2.** Activation of LPB^CCK neurons triggers aversion, defensive-like flight-to-nest behavior and autonomic responses. (**A**) Left, schematic of light stimulation of and patch-clamp recording from ChR2-EYFP expressing CCK neurons in the LPB. Right, example of action potentials evoked by optogenetic stimulation LPB^CCK neurons using whole cell patch-clamp slice recording. (**B**) Schematic diagram of optogenetic activation of LPB^CCK neurons. (**C**)

*Figure 2 continued on next page*

*Figure 2 continued*

Representative image showing the ChR2-EYFP expression and optical fiber tip locations in the LPB of a *Cck-cre* mouse. Scale bar, 100 μm. (**D**) Schematic of the timing and behavioral paradigm with optical activation of LPB[CCK] neurons. (**E–F**) Diagram of the real-time place aversion (RTPA) test and the example traces of the RTPA test from the mice. (**G**) Quantification of the time of mice spent in the laser-paired chamber (EYFP: n = 5 mice, ChR2: n = 5 mice; df = 16; two-way ANOVA test) after optogenetic activation. (**H**) Diagram of the flight to nest test. (**I–K**) Quantification of latency (**I**), speed (**J**) and time in the nest (**K**) (EYFP: n = 7 mice, ChR2: n = 7 mice; for latency, p = 0.0006, U = 0; Mann-Whitney test; for speed, p = 0.0016, t = 4.052, df = 12; unpaired *t* test; for time in the nest, p < 0.0001, t = 19.82, df = 12; unpaired *t* test). (**L**) Analyses of heart rate changes induced by photostimulation of LPB[CCK] neurons. (EYFP: n = 7 mice, ChR2: n = 7 mice; p = 0.0006, t = 4.603, df = 12; unpaired *t* test). (**M–N**) Example images of computer-detected pupils (**M**) and quantitative analyses of pupil size before and during photostimulation of LPB[CCK] neurons (**N**) (EYFP: n = 6 mice, ChR2: n = 6 mice; p = 0.0022, U = 0; Mann-Whitney test). (**O**) Cartoon of the arousal state during the activation of LPB[CCK] neurons. (**P**) Plasma corticosterone levels in EYFP and ChR2 groups. (EYFP: n = 5 mice, ChR2: n = 5 mice; p = 0.0121, t = 3.23, df = 8; unpaired *t* test).

The online version of this article includes the following source data and figure supplement(s) for figure 2:

**Source data 1.** Quantification of the flight-to-nest behavior and autonomic responses upon activation of LPB CCK neurons.

**Figure supplement 1.** Photostimulation of LPB[CCK] neurons induces anxiety-like behaviors.

**Figure supplement 1—source data 1.** Quantification of the anxiety-like behavior.

**Figure supplement 2.** Images of ChR2-EYFP expression in the LPB and optical fiber implantation above the LPB (**A-G**), with circles (**H-I**) indicating the location of optical fibers.

predatory cue stimulated the LPB[CCK] neurons (*Figure 3P–R*). Together, these data demonstrate that LPB[CCK] neurons encode innate threat stimuli-evoked aversion and flight behaviors.

## Inhibition of LPB[CCK] neurons suppressed predatory cue-evoked flight responses

To examine whether LPB[CCK] neurons are required for innate threat-evoked defensive behaviors, we then inhibited LPB[CCK] neurons using optogenetic tools. By injecting the Cre-dependent AAV expressing the guillardia theta anion channel rhodopsins-1 (GtACR1) into the LPB of *Cck-cre* mice, followed by optic stimulation, we were able to effectively inhibit the firing of LPB[CCK] neurons (*Figure 4A*). Next, we bilaterally injected the control or GtACR1 viruses into the LPB and photo-inhibited these neurons in vivo (*Figure 4B–C*). In the rat exposure test (*Figure 4D–E*), after a quick exploration of the rat in the corner (the danger zone), control mice usually fled to the other end of the box (the putative safe zone). However, optic inhibition of LPB[CCK] neurons significantly increased the time (EYFP, 7.247% ± 2.329% vs. GtACR1, 26.57% ± 5.375%) and entries (EYFP, 6.167 ± 1.579 vs. GtACR1, 12.67 ± 2.141) of mice to the danger zone, with no effect on total travel distance (EYFP, 8.837 ± 1.934 m vs. GtACR1, 12.69 ± 1.421 m), suggesting a delayed flight response (*Figure 4F–H*). In the looming test (*Figure 4I–J*), inhibition of LPB[CCK] neurons also increased the latency fleeing to the nest (EYFP, 10.18 ± 2.828 s vs. GtACR1, 27.14 ± 6.526 s), followed by reduced hiding time in the nest (EYFP, 85.36% ± 9.846% vs. GtACR1, 39.17% ± 16.36%; *Figure 4K–L*). Our data suggest that LPB[CCK] neurons are required for proper defensive behaviors to rat exposure and looming stimuli.

## Stimulation of the LPB[CCK]-PVN pathway triggers defensive-like flight-to-nest behaviors

To further investigate whether activation of the LPB[CCK]-PVN pathway induces defensive-like flight-to-nest behavior, we unilaterally injected the control or ChR2 virus into the LPB of *Cck-cre* mice and implanted an optical fiber above the PVN (*Figure 5A–E*). By photostimulating the axonal terminals of LPB[CCK] neurons within the PVN, we also observed shorter latency (EYFP, 73.43 ± 21.08 s vs. ChR2, 8 ± 1.047 s), faster speed (EYFP, 122.3% ± 17.81% vs. ChR2, 277.9% ± 32.85%) running towards the nest, and longer stay in the nest, (EYFP, 3.751% ± 1.23% vs. ChR2, 89.29% ± 10.71%; *Figure 5F–H*), similar to soma activation. Post-hoc c-fos staining further verified the activation of PVN neurons after optic stimulation (*Figure 5I–J*). As well, terminal activation also increased heart rates (*Figure 5K*) and plasma corticosterone levels (*Figure 5N*), but with no effects on pupil size (*Figure 5L–M*). Together,

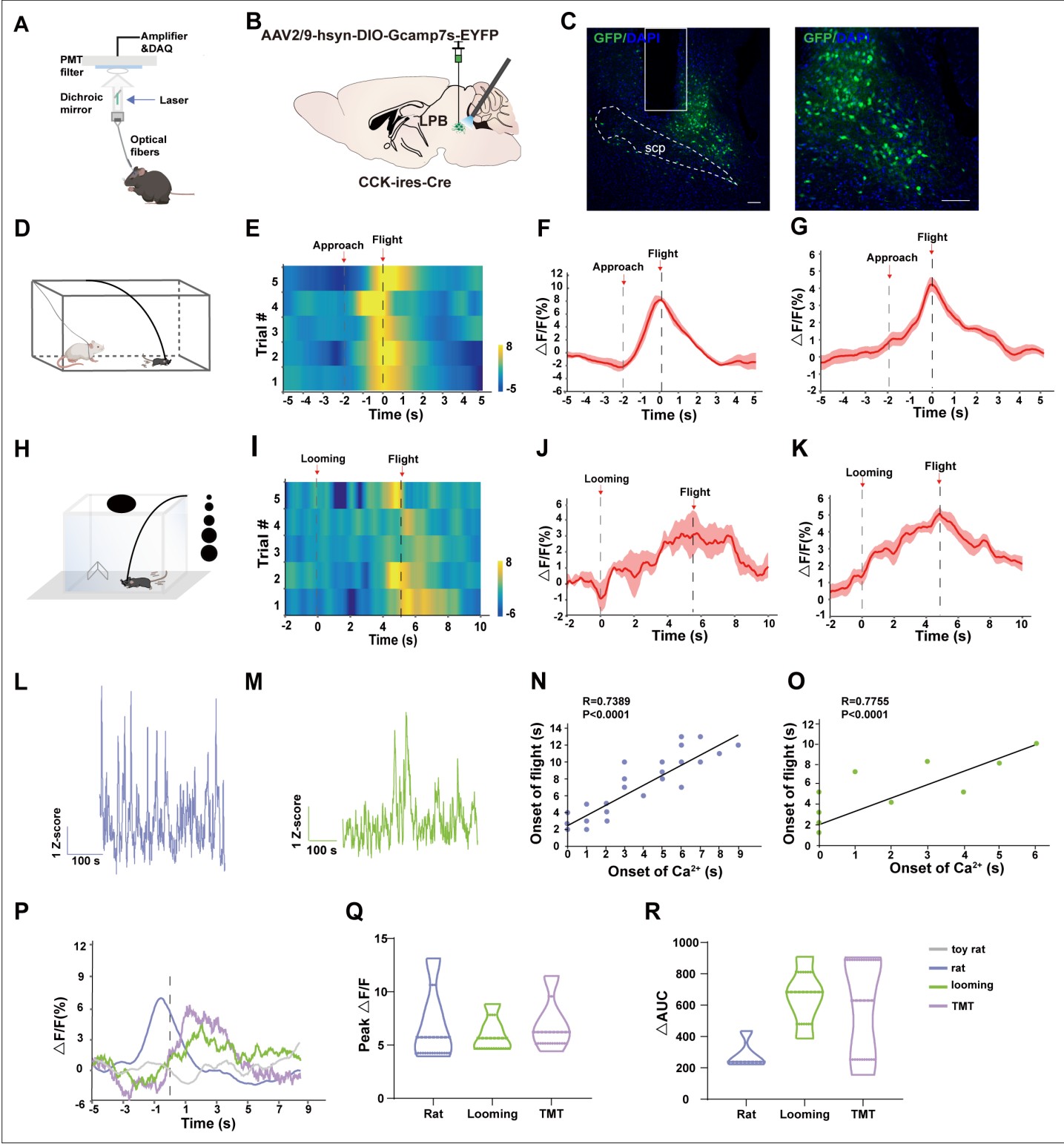

**Figure 3.** Threat stimuli recruit LPB[CCK] neurons to elicit defensive behaviors. (**A**) Schematic of the in vivo recording system for the Ca²⁺ signal. (**B**) Schematic showing the injections and recording of LPB neurons in *Cck-cre* mice. (**C**) A representative image (left panel) and a magnified view of neurons labeled by AAV-hsyn-DIO-GCaMP7s, scale bar, 100 μm; n = 5 mice. (**D**) Schematic of the rat exposure assay. (**E–F**) A heatmap (**E**) and a peri-event plot (**F**) of calcium transients of LPB[CCK] neurons in a mouse evoked by rat exposure (5 trials) (gray dotted line, onset of approach to the live rat; dark dotted line, onset of flight). (**G**) Average calcium transients of the tested animals during the rat exposure assay (n = 5 mice). Shaded areas around means indicate error bars. (**H**) Schematic paradigm of looming stimulus in a nest-containing open-field apparatus. (**I–J**) A heatmap presentation (**I**) and a peri-event plot

*Figure 3 continued on next page*

*Figure 3 continued*

(**J**) of calcium transients of LPB[CCK] neurons in a mouse upon looming stimulus (5 trials). (**K**) Average calcium transients of the tested animals during the looming assay (n = 5 mice). (**L–M**) Long session calcium recordings of LPB[CCK] neurons during rat exposure (**L**) or looming tests (**M**). (**N–O**) Correlation analyses between the elevation of calcium transients and the onset of flight during rat exposure or looming tests. (p < 0.0001 Linear regression). (**P**) Comparison of calcium signals LPB[CCK] neurons evoked by different stimuli. (**Q–R**) Plot depicting the differences of the amplitude or the area under the curve (AUC) of calcium signal changes in response to different stimuli. ΔAUC, AUC stimulus signal- AUC basal signal. (**Q**, Rat vs. Looming: p > 0.9999; Rat vs. TMT: p > 0.9999; TMT vs. Looming: p > 0.9999; one-way ANOVA; **R**, Rat vs. Looming p > 0.0676; Rat vs. TMT p > 0.1654; TMT vs. Looming p >0.9999; one-way ANOVA).

The online version of this article includes the following source data and figure supplement(s) for figure 3:

**Source data 1.** Plot depicting the difference of $Ca^{2+}$ activity.

**Figure supplement 1.** LPB[CCK] neurons respond to threat stimuli.

**Figure supplement 2. Images of AAV-DIO-synapse-jGCaMP7b virus infection at the LPB (A)** and downstream PVN and PSTh (**B-C**).

**Figure supplement 3. Images of Gcamp7s expression in the LPB and optical fiber implantation in the LPB (A-E)**, with circles (**F-G**) indicating the location of optical fibers.

these data suggest that activation of the LPB[CCK]-PVN pathway is sufficient to trigger flight-to-nest behavior, along with increased heart rates and corticosterone levels.

## PVN is required for defensive responses to threatening situations

Since LPB[CCK] neurons project to multiple regions (*Figure 1—figure supplement 2A–I*), the effects observed upon their activation may arise from different downstream targets, other than the PVN. To test this possibility, we investigated whether inhibition of the PVN is required for the LPB[CCK] neurons-induced defensive-like flight-to-nest behavior. To achieve this aim, we used a dual virus-mediated optogenetic activation and chemogenetic inhibition strategy, by injecting AAV2/9-DIO-ChR2-EYFP into the LPB, and AAV2/9-hsyn-hM4Di-mCherry into the PVN of *Cck-cre* mice. By putting an optical fiber above the PVN, we were able to activate the LPB[CCK] terminals and induce defensive-like flight-to-nest behavior as shown above (*Figure 6A–B*). Upon CNO administration to inhibit PVN neurons, we observed a much longer latency, lower speed of the mice in returning to the nest, and reduced hiding time in the nest (latency: saline, 8.286 ± 1.475 s vs. CNO, 18.29 ± 2.843 s; speed: saline, 289.3% ± 25.22% vs. CNO, 124.7% ± 13.67%; time in the nest: saline, 81.55% ± 11.16% vs. CNO, 25.83% ± 10.32%; *Figure 6C–E*). These data suggest that the PVN is a required downstream target for LPB[CCK] neurons activation-induced defensive behavior.

LPB[CCK] neurons may release either glutamate or neuropeptide CCK to induce the defensive behavior. To determine which neurotransmitter or modulator is engaged in the above responses, we combined pharmacologic with optogenetic manipulation. Two types of CCK receptors are present in the central nervous system and *Cckar* is widely distributed throughout the PVN, whereas the expression of *Cckbr* in PVN is low (*Figure 6—figure supplement 1A–B*). We infused glutamate receptor antagonists (CNQX +AP5), CCK receptor antagonists (Devazepide +L-365,260), or artificial cerebrospinal fluid (ACSF) through an implanted cannula 30 min before optogenetic activation of LPB[CCK] neurons (*Figure 6F–H*). We found that blocking glutamate receptors induced a longer latency, slower speed in returning to the nest, along with reduced hiding time in the nest (*Figure 6I–K*), when compared with ACSF-treated mice. However, blocking CCK receptors showed no significant effects (*Figure 6I–K*). These results suggest that glutamatergic transmission from LPB[CCK] neurons to the PVN mediates defensive-like flight-to-nest behavior.

## Photostimulation of PVN[Vglut2] neurons promotes defensive-like flight-to-nest behavior

Of note, PVN neurons are mostly glutamatergic (*Vong et al., 2011*; *Xu et al., 2013*). We further assessed whether optogenetic activation of PVN[Vglut2] neurons elicit defensive-like behavior (*Figure 6L–M*). We found that optogenetic activation of PVN[Vglut2] neurons also induced flight-to-nest behavior (latency: EYFP, 127.5 ± 22.5 s vs. ChR2, 12.83 ± 2.937 s; speed: EYFP, 142.4 ± 32.07% vs. ChR2, 384.4% ± 83.52%), followed by longer hiding time in the nest (EYFP, 0.2783% ± 0.2783% vs. ChR2, 61.11% ± 17.58%) (*Figure 6N–P*), compared to the control. These data suggest that optogenetic activation of PVN[Vglut2] neurons was sufficient to evoke defensive-like flight-to-nest behavior.

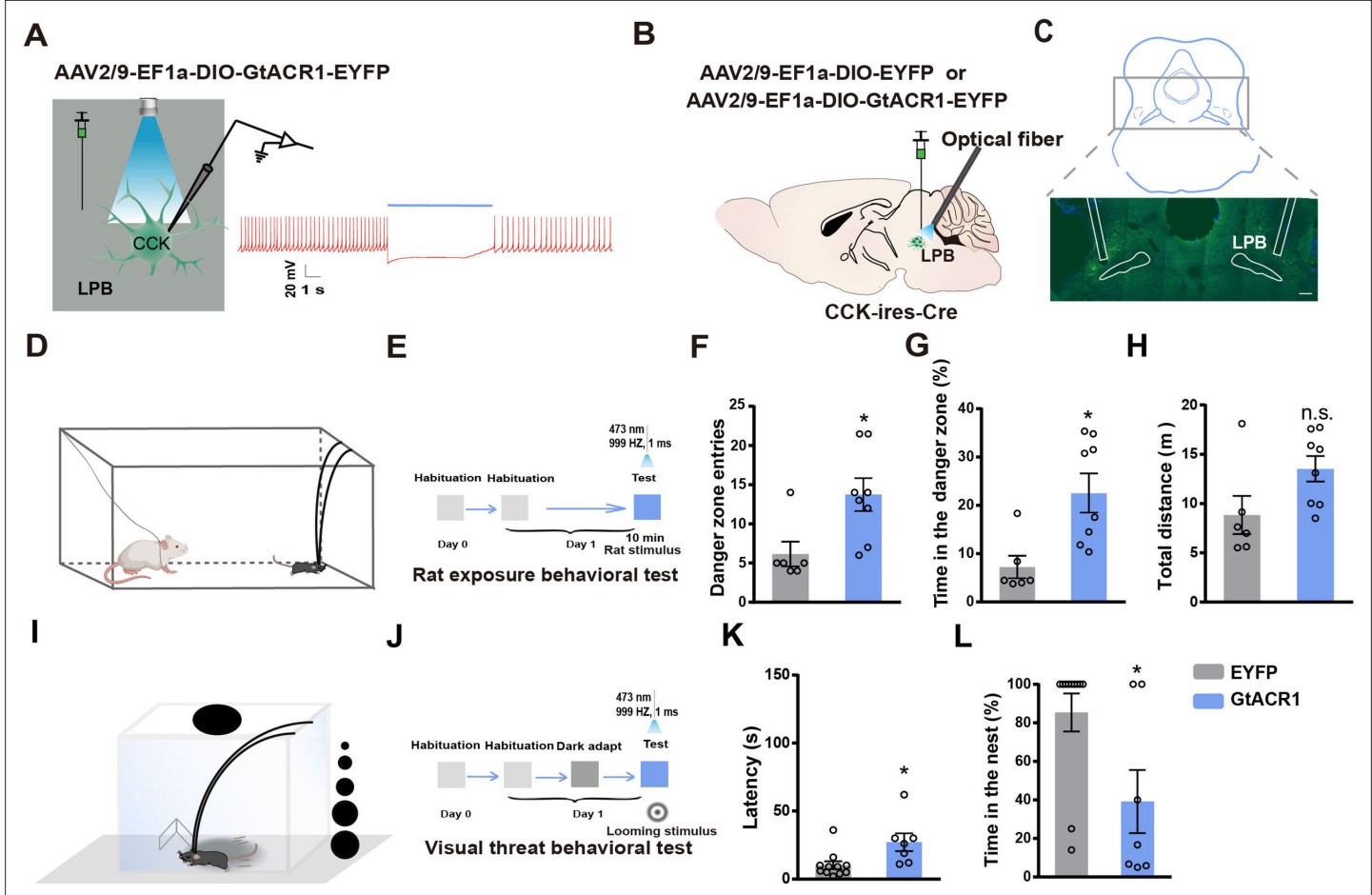

**Figure 4.** Optogenetic inhibition of LPB^CCK neurons suppresses predator- and visual predatory cue-evoked innate flight responses. (**A**) Left, schematic of light stimulation of and patch-clamp recording of GtACR1-expressing CCK neurons in the LPB. Right, example of action potentials evoked by optogenetic inhibition of LPB^CCK neurons from using whole cell patch-clamp recording. (**B**) Schematic diagram of optogenetic inhibition of LPB^CCK neurons. (**C**) Representative image showing the GtACR1-EYFP expression in the LPB and optical fiber tip locations above the LPB of a *Cck-cre* mouse. (**D–E**) Schematic and the timing and behavioral paradigm of rat exposure assay. (**F–H**) Photoinhibition of LPB^CCK neurons increased number of entries toward the rat (danger zone), time spent in the danger zone, with unchanged travel distance (EYFP: n = 6 mice, GtACR1: n = 9 mice; for times of entries, p = 0.0456, t = 2.21, df = 13; unpaired *t* test; for time in the danger zone, p = 0.0153, t = 2.791, df = 13; unpaired *t* test; for total distance, p = 0.1246, t = 1.642, df = 13; unpaired *t* test). (**I–J**) Schematic of the looming test apparatus, the timing and behavioral paradigm of looming-evoked flight-to-nest behavioral test. (**K–L**) Photoinhibition of LPB^CCK neurons increased the latency towards the nest and reduced the hiding time in the nest. (EYFP: n = 11 mice, GtACR1: n = 7 mice; for latency, p = 0.0153, t = 2.716, df = 16; unpaired *t* test; Mann-Whitney test; for time in the nest, p = 0.0201, t = 2.582, df = 16; unpaired *t* test).

The online version of this article includes the following source data and figure supplement(s) for figure 4:

**Source data 1.** Quantification of the defensive responses upon inhibition of LPB CCK neurons.

**Figure supplement 1.** Images of GtACR1-EYFP expression in the LPB and optical fiber implantation above the LPB (**A-I**), with circles (**J-L**) indicating the location of optical fibers.

PVN^CRH neurons have recently been shown to predict the occurrence of defensive behaviors in mice (*Daviu et al., 2020*). Interestingly, optogenetic activation of the LPB^CCK-PVN pathway primarily activated PVN^CRH neurons, rather than vasopressin or oxytocin neurons (*Figure 6—figure supplement 1C–F*). However, optogenetic activation of PVN^CRH neurons was not sufficient to drive defensive-like flight-to-nest behavior (*Figure 6—figure supplement 1G–K*; latency: EYFP, 77 ± 20.69 s vs. ChR2, 90.17 ± 22.03 s; speed: EYFP, 156.6 ± 22.59% vs. ChR2, 217.2% ± 61.54%; time in the nest: EYFP, 20.95% ± 13.89% vs. ChR2, 35.28% ± 20.48%).

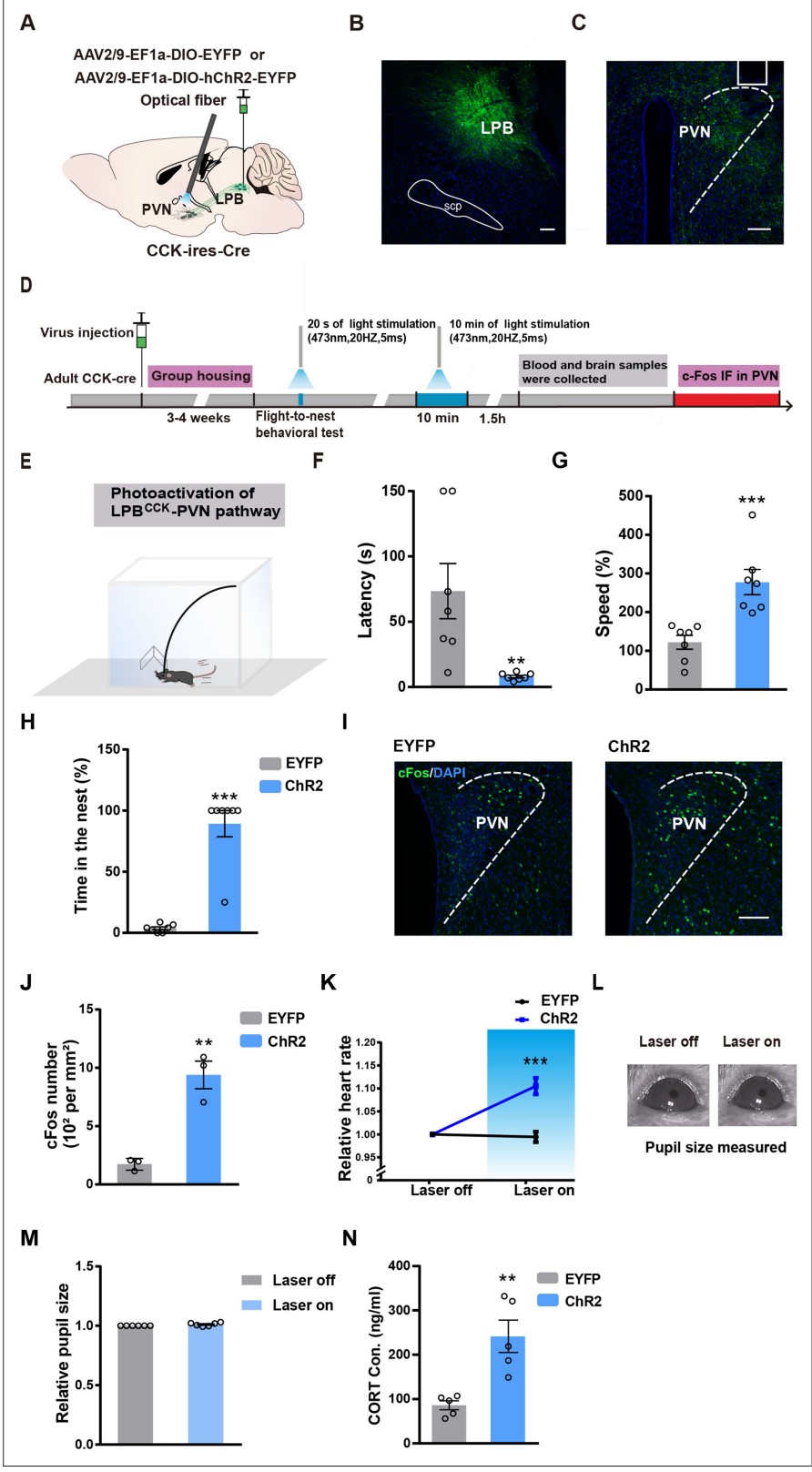

**Figure 5.** Photostimulation of the LPB^CCK-PVN pathway induces defensive-like flight-to-nest behavior. (**A**) Schematic diagram of optogenetic activation of LPB^CCK-PVN pathway. (**B–C**) A representative image showing the ChR2-EYFP expression in the LPB (**B**) and the optical fiber tip locations in the PVN (**C**) of a *Cck-cre* mouse. (**D**) Schematic of the timing and behavioral paradigm with optical activation of LPB^CCK-PVN pathway. (**E**) Schematic

*Figure 5 continued on next page*

*Figure 5 continued*

of the experimental apparatus with a nest in the corner. (**F–H**) Optogenetic activation of the LPB$^{CCK}$-PVN pathway shortened the latency but increased the speed of animals towards the nest, with increased hiding time in the nest (EYFP: n = 7 mice, ChR2: n = 7 mice; for latency, p = 0.0012, U = 0; Mann-Whitney test; for speed, p = 0.0013, t = 4.163, df = 12; unpaired *t* test; for time in the nest, p = 0.0006, U = 0; Mann-Whitney test). (**I–J**) C-fos staining in the PVN (**I**) and quantification of c-fos positive cells in the PVN (**J**). Scale bar: 100 µm. (EYFP: n = 3 mice, ChR2: n = 3 mice; p = 0.0033, t = 6.253, df = 4; unpaired *t* test). (**K**) Mean heart rate analyses in EYFP and ChR2 groups (EYFP: n = 7 mice, ChR2: n = 7 mice; p = 0.0002, t = 5.249, df = 12; unpaired *t* test). (**L**) Example image of computer-detected pupil size before and during photoactivation of LPB$^{CCK}$ neurons. (**M**) Relative pupil size of animals (during/ before photostimulation of LPB$^{CCK}$ neurons) (EYFP: n = 6 mice, ChR2: n = 6 mice; p = 0.0022, U = 0; Mann-Whitney test). (**N**) Plasma corticosterone levels in EYFP and ChR2 groups (EYFP: n = 5 mice, ChR2: n = 5 mice; p = 0.0079, U = 0; Mann-Whitney test).

The online version of this article includes the following source data and figure supplement(s) for figure 5:

**Source data 1.** Quantification of the flight-to-nest behavior and autonomic responses upon activation of LPB CCK-PVN pathway.

**Figure supplement 1.** Images of ChR2-EYFP expression in the LPB and optical fiber implantation above the PVN (**A-F**).

## Discussion

Defensive behaviors are actions that are naturally selected to avoid or reduce potential harm for the survival of animal species. While emerging evidence has suggested that LPB are associated with sensation of danger signals and aversive stimuli (*Campos et al., 2018*; *Han et al., 2015b*), we found that LPB$^{CCK}$ neurons were actively recruited upon exposure to various predatory stimuli. Activating LPB$^{CCK}$ neurons triggers aversive-like avoidance and defensive-like flight responses, whereas inhibition of these neurons suppressed predatory stimuli-induced defensive behaviors. Inhibition of downstream PVN neurons attenuated photoactivation of LPB$^{CCK}$-PVN terminals-promoted flight responses, suggesting that LPB$^{CCK}$-PVN pathway is important for the regulation of defensive responses. Our study thus reveals a new connection from the brainstem to the hypothalamus to regulate innate defensive behaviors (*Figure 6Q*).

Hippocampal and cortical CCK-expressing neurons have been thought of mainly GABAergic (*Liu et al., 2020*; *Sun et al., 2020a*; *Whissell et al., 2015*). We found that LPB$^{CCK}$ neurons are predominantly glutamatergic, similar to those in the amygdala (*Shen et al., 2019*). While the roles of LPB$^{CGRP}$ neurons have been associated with the passive defensive responses (*Han et al., 2015b*; *Keay and Bandler, 2001*), we show that activation of LPB$^{CCK}$ neurons primarily triggers active defensive responses, such as flight-to-nest behavior (*Keay and Bandler, 2001*). The fact that CGRP and CCK neurons are spatially segregated, responsive to distinct threat signals and involved in different defensive responses, indicating that LPB acts as an important integration center or hub to gate defensive responses when animals encounter different threats (*Liu et al., 2022*; *Tokita et al., 2009*). It would be interesting to investigate whether LPB$^{CCK}$ and LPB$^{CGRP}$ neurons coordinately regulate defensive responses under different threatening situations.

In vivo calcium imaging showed that LPB$^{CCK}$ neurons increase firing upon exposure to predators and/or predatory cues, which occurs before the initiation of an escape behavior. These findings indicate that LPB$^{CCK}$ neurons may produce a preparatory signal before the escape initiation, which allows the downstream targets to further assess the threat signals, plan escape routes and take appropriate motor actions. LPB$^{CCK}$ neurons are well positioned to play such an important role in linking the imminent threat to escape initiation, as they receive inputs from both the peripheral and visceral sensory system and project to many brain regions involved in defensive responses including the hypothalamus, PVT and PAG (*Cooper and Blumstein, 2015*; *Ellard and Eller, 2009*; *Han et al., 2015b*; *Krout and Loewy, 2000*; *Saper and Loewy, 1980*; *Sun et al., 2020b*; *Tokita et al., 2009*). While PVN$^{CRH}$ or PAG$^{CCK}$ neurons have also been shown to be recruited during flight, their peak activity did not match the flight initiation, but occurred during flight (*Daviu et al., 2020*; *La-Vu et al., 2022*). The peak

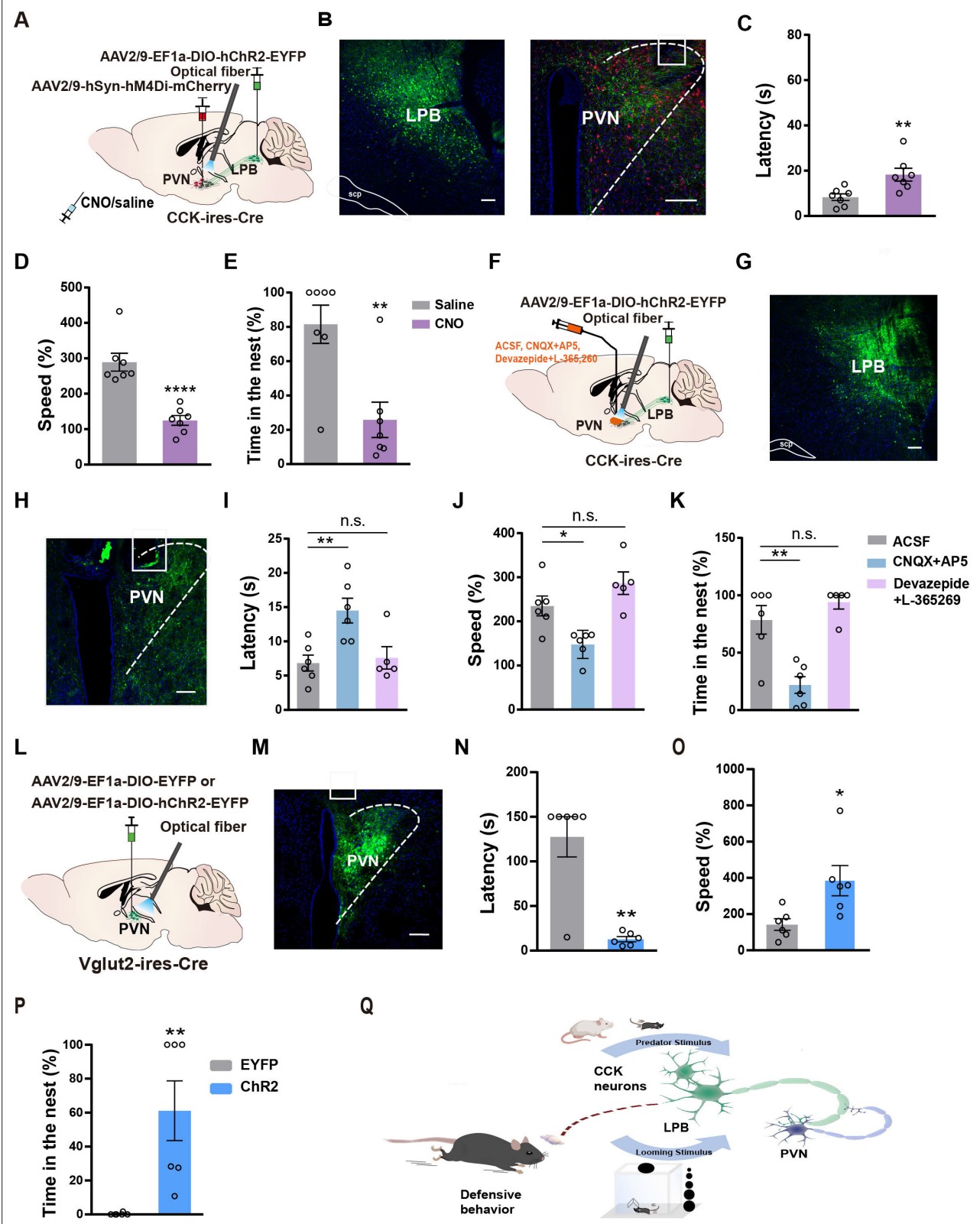

**Figure 6.** PVN is involved in the defensive-like flight-to-nest behavior evoked by LPB^CCK neurons. (**A**) Schematic of experimental setup. (**B**) Left, representative image showing the ChR2-EYFP expression in the LPB of a *Cck-cre* mouse. Scale bar, 100 μm. Right, representative image showing the hM4Di-mCherry expression in the PVN and optic fiber placement in the PVN from a ChR2-EYFP expressing mouse. (**C–E**) Chemogenetic inhibition of PVN neurons before optical activation of LPB^CCK-PVN terminals increased latency and reduced the speed of animals towards the nest, with reduced

*Figure 6 continued on next page*

*Figure 6 continued*

hiding time in the nest (saline: n = 7 mice, CNO: n = 7 mice; for latency, p = 0.0088, t = 3.122, df = 12; unpaired *t* test; for speed, p < 0.0001, t = 5.736, df = 12; unpaired *t* test; for time in the nest, p = 0.0032, t = 3.665, df = 12; unpaired *t* test). (**F**) Optogenetic activation of LPB$^{CCK}$-PVN terminals with a cannula implanted in the PVN for aCSF, CNQX +AP5, or Devazepide +L-365 260 delivery. Scale bar, 100 µm. (**G**) Representative image showing the ChR2-EYFP expression in the LPB of a *Cck-cre* mouse. Scale bar, 100 µm. (**H**) Representative image showing the cannula placement in the PVN from a ChR2-EYFP expressing *Cck-cre* mouse. Scale bar, 100 µm. (**I–K**) Microinjection of the glutamate receptor antagonists (CNQX +AP5), rather than CCK receptor antagonists (Devazepide +L-365260), into the PVN increased the latency to the nest and reduced the hiding time in the nest (ACSF: n = 6 mice, CNQX +AP5: n = 6 mice, Devazepide +L-365260: n = 5 mice; for latency, F(2,14) = 7.658, p = 0.057; One-way ANOVA; for speed, F(2,14) = 11.49, p = 0.0011; for time in the nest, One-way ANOVA; F(2,14) = 16.44, p = 0.0002; One-way ANOVA). (**L**) Schematic diagram of optogenetic activation of PVN$^{Vglut2}$ neurons. (**M**) Representative image showing the ChR2-EYFP expression and optical fiber tip locations in the PVN of a *Vglut2-cre* mouse. Scale bar, 100 µm. (**N–P**) Optogenetic activation of PVN$^{Vglut2}$ neurons reduced the latency, increased the speed of animals towards a nest and the time in the nest. (EYFP: n = 6 mice, ChR2: n = 6 mice; for latency, p = 0.0065, U = 2; Mann-Whitney test; for speed, p = 0.0221, t = 2.705, df = 10; unpaired *t* test; for time in the nest, p = 0.022, U = 0; Mann-Whitney test). (**Q**) Graphical summary showing the LPB$^{CCK}$-PVN pathway in mediating defensive behaviors.

The online version of this article includes the following source data and figure supplement(s) for figure 6:

**Source data 1.** Quantification of the flight-to-nest behavior upon manipulation of PVN neurons.

**Source data 2.** Maps for virus expression and optical fiber location.

**Figure supplement 1.** Stimulation of LPB$^{CCK}$-PVN pathway activates PVN$^{CRH}$ neurons, but activation of PVN$^{CRH}$ neurons does not trigger flight-to-nest behavior.

**Figure supplement 1—source data 1.** Quantification of the CRH neurons and flight-to-nest behavior.

**Figure supplement 2.** Images of ISH data from the Allen Brain Atlas.

**Figure supplement 3.** Images of ChR2-EYFP expression in the PVN and optical fiber implantation above the PVN (**A-F**).

activities of LPB$^{CCK}$ neurons at the onset of escape suggest that LPB$^{CCK}$ neurons likely gate and instruct the escape rather than directly control locomotive behaviors.

Despite many early studies mapping downstream projections for LPB neurons (*Han et al., 2015a*; *Saper and Loewy, 1980*; *Sun et al., 2020b*), we are the first to demonstrate that LPB$^{CCK}$ neurons directly connect with PVN neurons. Direct projections from the brainstem to the hypothalamus for the escape responses may be more conducive to escape in emergency situations. Besides the PVN, we also observed that LPB$^{CCK}$ neurons project to the PVT, VMH and PAG, which have also been associated with aversion and/or defensive behaviors (*LeDoux, 2012*; *Vianna et al., 2001*; *Wang et al., 2015*). It would be interesting to further investigate how these different downstream areas coordinately contribute to defensive responses.

PVN is important for neuroendocrine and autonomous regulation (*Canteras et al., 2001*; *Coote, 2005*; *Ferguson et al., 2008*; *Sutton et al., 2016*). While earlier c-fos staining analyses suggest that PVN neurons are activated in response to different threat stimuli (*Canteras et al., 2001*; *Faturi et al., 2014*; *Martinez et al., 2008*; *Staples et al., 2008*), it remains unclear whether they are only involved in neuroendocrine responses or also defensive behaviors. Increasing evidence has demonstrated that PVN neurons play important roles in defensive behaviors independent of hormonal actions (*Daviu et al., 2020*; *Mangieri et al., 2019*; *Xu et al., 2019*). Sim1, as a specific marker of PVN, studies have reported that photoactivation of PVN$^{Sim1}$ neurons induced defensive responses (*Mangieri et al., 2019*). Our findings support a role for PVN neurons in mediating behavioral responses, as photoactivation of PVN$^{Vglut2}$ neurons promoted flight-to-nest behavior. Intriguingly, stimulating the LPB$^{CCK}$-PVN pathway promoted flight-to-nest behaviors but activating of PVN$^{CRH}$ neurons did not induce apparent flight-to-nest behaviors. Since PVN$^{CRH}$ neurons receive multiple inputs and activation of the PVN$^{CRH}$ neurons have been shown to induce aversion (*Kim et al., 2019*), future studies using input-specific activation of PVN$^{CRH}$ neuronal subpopulations would further dissect the role of LPB$^{CCK}$- PVN$^{CRH}$ pathway in defensive responses. Future studies using additional projection-specific approaches and more genetically defined cell tools may help resolve this problem.

Our data show that activation of LPB$^{CCK}$ neurons drives aversion, defensive and anxiety-like behaviors, evoking an arousal and stressed states. Fear, anxiety and stress responses are closely associated with mental illnesses with dysregulated neural circuits (*Blanchard et al., 2001*). It is worth further

investigation to examine whether alterations of the LPB^CCK-PVN pathway are implicated in these diseases and whether manipulation of this pathway might be an effective strategy for therapeutic targeting.

# Materials and methods

## Key resources table

| Reagent type (species) or resource | Designation | Source or reference | Identifiers | Additional information |
|---|---|---|---|---|
| Strain, strain background (*Mus musculus*) | *Cck-ires-cre(Cck^{tm1.1(cre)Zjh}/J)* | The Jackson Laboratory | JAX012706 | |
| Strain, strain background (*Mus musculus*) | *Crh-ires-cre (B6(Cg))-Crh^{tm1(cre)Zjh}/J* | The Jackson Laboratory | JAX012704 | |
| Strain, strain background (*Mus musculus*) | *Vglut2-ires-Cre (Slc17a6^{tm2(cre)Lowl}/J)* | The Jackson Laboratory | JAX 016963 | |
| Strain, strain background (*Mus musculus*) | *Ai14 (B6;129S6-Gt(ROSA)26Sor^{tm14(CAG-tdTomato)Hze}/J)* | The Jackson Laboratory | JAX007908 | |
| Strain, strain background (*Rattus norvegicus*) | SD | Shanghai SLAC Laboratory Animal Co. Ltd | http://www.slaccas.com/ | |
| Strain, strain background (*Mus musculus*) | C57BL/6 J | Shanghai SLAC Laboratory Animal Co. Ltd | http://www.slaccas.com/ | |
| Antibody | Anti-Dsred, rabbit polyclonal | Takara | Cat# 632496 RRID: AB_10013483 | (1:800) |
| Antibody | Anti-CGRP, mouse monoclonal | Abcam | Cat# 81887 RRID: AB_1658411 | (1:800) |
| Antibody | Anti-GFP, goat polyclonal | Abcam | Cat# 5450 RRID: AB_304896 | (1:500) |
| Antibody | Anti-CRH, rabbit polyclonal | Phoenix Biotech | Cat# H-019–06 | (1:500) |
| Antibody | Anti-c-fos, Guinea pig polyclonal | Synaptic Systems | Cat# 226004 RRID: AB_2619946 | (1:10,000) |
| Antibody | Alexa Fluor 488 donkey anti-guinea pig IgG (H+L) polyclonal | Jackson | Cat#112-486-068 RRID: AB_2617153 | (1:1000) |
| Antibody | Alexa Fluor 555 donkey anti-rabbit IgG (H+L) polyclonal | Invitrogen | Cat#A31572 RRID: AB_162543 | (1:1000) |
| Antibody | Alexa Fluor 488 donkey anti-mouse IgG (H+L) polyclonal | Invitrogen | Cat# R37114 RRID: AB_2556542 | (1:1000) |
| Antibody | Alexa Fluor 488 donkey anti-goat IgG (H+L) polyclonal | Invitrogen | Cat# A11055 RRID: AB_2534102 | (1:1000) |
| Antibody | Alexa Fluor 488 donkey anti-rabbit IgG (H+L) polyclonal | Jackson | Cat# R37118 RRID: AB_2556546 | (1:1000) |
| Commercial assay or kit | RNAscope Multiplex Fluorescent Reagent Kit v2 | Advanced Cell Diagnostics | Cat# 323100 | |
| Sequence-based reagent | RNAscope probe *Slc17a6* | Advanced Cell Diagnostics | accession number NM_080853.3 | probe region 1986–2998 |
| Sequence-based reagent | RNAscope probe *Slc32a1* | Advanced Cell Diagnostics | Accession number NM_009508.2 | probe region 894–2037 |
| Sequence-based reagent | RNAscope probe *Cckar* | Advanced Cell Diagnostics | accession number NM_009827.2 | probe region 328–1434 |
| Sequence-based reagent | RNAscope probe *Cckbr* | Advanced Cell Diagnostics | accession number NM_007627.4 | probe region 136–1164 |

*Continued on next page*

Continued

| Reagent type (species) or resource | Designation | Source or reference | Identifiers | Additional information |
|---|---|---|---|---|
| Chemical compound, drug | CNQX disodium salt hydrate | Sigma-Aldrich | Cat#1045 | |
| Chemical compound, drug | Devazepide | Sigma-Aldrich | Cat#2304 | |
| Chemical compound, drug | Clozapine N-oxide | Sigma-Aldrich | Cat#C0832 | |
| Chemical compound, drug | L-365,260 | Sigma-Aldrich | Cat#143626 | |
| Chemical compound, drug | DAPI | Sigma-Aldrich | N/A | |
| Chemical compound, drug | Tween-20 | Sigma-Aldrich | N/A | |
| Strain, strain background (AAV2/9) | AAV2/9-hEF1a-DIO-hChR2(H134R)-EYFP-WPRE-pA | Shanghai Taitool Bioscience Co. | Cat# S0199-9 | Viral titers: $2.95 \times 10^{13}$ particles/ml |
| Strain, strain background (AAV2/9) | AAV2/9-hEF1a-DIO-EYFP-WPRE-pA | Shanghai Taitool Bioscience Co. | Cat#S0196-9 | Viral titers: $1.0 \times 10^{12}$ particles/ml |
| Strain, strain background (AAV2/9) | AAV2/9-CAG-DIO-hGtACR1-P2A-EGFP-WPRE-pA | Shanghai Taitool Bioscience Co. | Cat# S0311-9 | Viral titers: $5 \times 10^{13}$ particles/ml |
| Strain, strain background (AAV2/9) | rAAV2/9-hSyn-DIO-mGFP-2A-Synaptophysin-mRuby | Shanghai Taitool Bioscience Co. | Cat# S0250-9 | Viral titers: $1.55 \times 10^{13}$ particles/ml |
| Strain, strain background (AAV2/2) | rAAV2/2-Retro-hEF1a-DIO-EYFP-WPRE-pA | Shanghai Taitool Bioscience Co. | Cat# S0196-2R | Viral titers: $2.52 \times 10^{13}$ particles/ml |
| Strain, strain background (AAV2/9) | AAV2/9-hSyn-mCherry-WPRE-pA | Shanghai Taitool Bioscience Co. | Cat# S0238-9 | Viral titers: $\geq 1.0 \times 10^{13}$ particles/ml |
| Strain, strain background (AAV2/9) | AAV2/9-hSyn-hM4D(Gi)-mCherry-WPRE-pA | Shanghai Taitool Bioscience Co. | Cat# S0279-9 | Viral titers: $\geq 1.0 \times 10^{13}$ particles/ml |
| Strain, strain background (AAV2/9) | AAV2/9-hsyn-DIO-jGCaMP7s-WPRE-pA | Shanghai Taitool Bioscience Co. | Cat# S0590-9 | Viral titers: $\geq 1.0 \times 10^{13}$ particles/ml |
| Software, algorithm | ANY-Maze software 5.3 | Global Biotech Inc | http://www.anymaze.co.uk/ | |
| Software, algorithm | Image J | NIH | https://imagej.nih.gov/ij/index.html;%20 RRID:SCR_003070 | |
| Software, algorithm | GraphPad Prism 6 | GraphPad Software | https://www.graphpad.com/scientificsoftware/prism/; RRID: SCR_002798 | |
| Software, algorithm | MatLab R2016a | MathWorks | https://www.mathworks.com/products.html; RRID:SCR_001622 | |

## Animals

*Cck-ires-cre* (*Cck*[tm1.1(cre)Zjh]/J; Stock No. 012706), *Crh-ires-cre* (*B6(Cg)-Crh*[tm1(cre)Zjh]/J; Stock No. 012704), *Vglut2-ires-cre* (*Slc17a6*[tm2(cre)Lowl]/J; Stock No. 016963), *Ai14* (*B6;129S6-Gt(ROSA)26Sor*[tm14(CAG-tdTomato)Hze]/J; Stock No. 007908), C57BL/6 mice and SD rats were obtained from the Shanghai Laboratory Animal Center. Adult male mice and SD rats were used in our study. Mice and rats housed at 22 ± 1 °C and 55 ± 5% humidity on a 12 hr light/12 hr dark cycle (light on from 07:00 to 19:00) with food and water ad libitum. All experimental procedures were approved by the Animal Advisory Committee at Zhejiang University and were performed in strict accordance with the National Institutes of Health Guidelines for the Care and Use of Laboratory Animals. All surgeries were performed under sodium pentobarbital anesthesia, and every effort was made to minimize suffering.

## Immunohistochemistry

Animals were transcardially perfused with saline and 4% PFA. Brains were post-fixed overnight in 4% PFA at 4 °C, followed by immersed in 30% sucrose solution. Coronal sections (40 µm) were cut by a CM1950 Microtome (Leica). Immunostaining was performed as previously described (*Zhang et al., 2018*). The brain slices were permeabilized in 0.5% Triton X-100 in Tris-buffered saline, blocked with 100 mM glycine and 5% bovine serum albumin (BSA) containing 5% normal donkey serum. Tissue sections were subsequently incubated with diluted primary and secondary antibodies as indicated, nuclei stained with 6-diamidino2-phenylindole (DAPI), and slides mounted with antifade reagents. The primary antibodies used were: Guinea pig anti-c-fos (Synaptic Systems, Cat# 226004), Rabbit anti-Dsred (Takara, Cat# 632496), Goat anti-GFP (Abcam, Cat# 5450), Mouse anti-CGRP (Abcam, Cat# 81887). Slides were imaged with a confocal microscope (Olympus FluoView FV1200).

## RNAscope in situ hybridization

We used RNAscope multiplex fluorescent reagent kit and designed probes (ACDBio Inc) to perform fluorescence in situ hybridization. Mouse brain tissue was sectioned into 20 µm sections by cryostat (Leica CM 1950). Then sections were mounted on slides and air-dried at room temperature. Subsequently, the sections were dehydrated in 50% EtOH, 70% EtOH, and 100% EtOH for 5–10 min each time and air-dried at room temperature again. Thereafter, protease digestion was performed in a 40 °C HybEZ oven for 30 min pretreatment, slides were hybridized with pre-warmed probe in a 40 °C HybEZ oven for 2 hr. Probes used in our paper were: *Slc17a6* probe (*VGLUT2*, accession number NM_080853.3, probe region 1986–2998), *Slc32a1* probe (*VGAT*, accession number NM_009508.2, probe region 894–2037), *Cckar* probe (accession number NM_009827.2, probe region 328–1434), *Cckbr* probe (accession number NM_007627.4, probe region 136–1164). After hybridization, the brain sections went through four steps of signal amplification fluorescent label. Anti-DsRed or GFP staining was performed after the RNAscope staining. Slides were imaged with a confocal microscope (Olympus FluoView FV1200).

## Stereotaxic injections and optical fiber/cannula implantation

For surgical procedures, mice were anaesthetized with sodium pentobarbital (0.1 g/kg) and placed in a stereotaxic apparatus (RWD). Stereotaxic surgery was performed as described. Briefly, holes were made into the skull over the target areas to inject the virus with glass pipettes (diameter 10–15 mm) or to implant optical fibers (outer diameter [o.d.]: 200 µm; length: 6.0 mm, 0.37 NA; Inper) or to implant of guide cannula (outer diameter [o.d.]: 0.41 mm; RWD). The coordinates relative to bregma were as follows according to the Paxinos and Franklin (2001) atlas. For all experiments, mice with incorrect injection sites were excluded from further analysis.

For optical activation of LPB^CCK neurons, AAV2/9-hEF1a-DIO-hChR2(H134R)-EYFP-WPRE-pA (viral titers: $2.95\times10^{13}$ particles/ml; Taitool Bioscience) or AAV2/9-hEF1a-DIO-EYFP-WPRE-pA (viral titers: $1.0\times10^{12}$ particles/ml; Taitool Bioscience) virus was unilaterally microinjected into the LPB (AP: –4.8; ML: –1.35; DV: –3.4; mm relative to bregma) of *Cck-cre* mice. The virus was diluted into $5.9\times10^{12}$ genomic copies per ml with phosphate-buffered saline (PBS) before use and injected with 65 nl into the LPB. Virus was delivered at a flow rate of 10 nl/min. The glass capillary was left in place for an additional 10 min after injection to allow diffusion of the virus. The cannulas were held in place with dental cement above the LPB (AP: –4.8; ML: –1.35; DV: –3.2; mm relative to bregma).

For optical activation of PVN^Vglut2 neurons, AAV2/9-EF1a-DIO-hChR2-EYFP (viral titers: $2.95\times10^{13}$ particles/ml; Taitool Bioscience) or AAV2/9-EF1α-DIO-EYFP (viral titers: $1.0\times10^{12}$ particles/ml; Taitool Bioscience) virus was unilaterally microinjected into the PVN (AP: –0.4; ML: –0.15; DV: –4.95; mm relative to bregma) of *Vglut2-cre* mice. The virus was diluted into $2.0\times10^{12}$ genomic copies per ml with PBS before use and injected with 70 nl into the PVN. Virus was delivered at a flow rate of 15 nl/min. The glass capillary was left in place for an additional 10 min after injection to allow diffusion of the virus. The cannulas were held in place with dental cement above the PVN (AP: –0.4; ML: –0.15; DV: –4.7; mm relative to bregma).

For optical activation of PVN^CRH neurons, AAV2/9-EF1a-DIO-hChR2-EYFP (viral titers: $2.95\times10^{13}$ particles/ml; Taitool Bioscience) or AAV2/9-EF1α-DIO-EYFP (viral titers: $1.0\times10^{12}$ particles/ml; Taitool Bioscience) was unilaterally microinjected into the PVN (AP: –0.4; ML: –0.15; DV: –4.95 mm relative to bregma) of *Crh-cre* mice. The virus was diluted into $4.0\times10^{12}$ genomic copies per ml with PBS before

use and injected with 75 nl into the PVN. Virus was delivered at a flow rate of 15 nl/min. The glass capillary was left in place for an additional 10 min after injection to allow diffusion of the virus. The cannulas were held in place with dental cement above the PVN (AP: –0.4; ML: –0.15; DV: –4.7; mm relative to bregma).

For optical inhibition of LPB$^{CCK}$ neurons, AAV2/9-EF1α-DIO-hGtACR1-P2A-eYFP-WPRE (viral titers: 5×10$^{13}$ particles/ml; Taitool Bioscience) or AAV2/9-hEF1a-DIO-EYFP-WPRE-pA (viral titers: 1.0×10$^{12}$ particles/ml; Taitool Bioscience) virus was bilaterally infused to the LPB (AP: –4.8; ML: –1.35; DV: –3.4; mm relative to bregma) of *Cck-cre* mice. The virus was diluted into 1.67×10$^{12}$ genomic copies per ml with PBS before use and injected with 70 nl into the LPB. Optical fibers were bilaterally implanted at an angle of 5° above the LPB (AP: –4.8; ML: –1.72; DV: –2.94; mm relative to bregma).

In order to study the downstream of LPB$^{CCK}$ neurons, AAV2/9-hSyn-DIO-mGFP-2A-Synaptophysin-mRuby (viral titers: 1.55×10$^{13}$ particles/ml; Taitool Bioscience) virus was unilaterally injected into the LPB (AP: –4.8 mm; ML: –1.35 mm; DV: –3.4 mm) of *Cck-cre* mice; in order to study the source of CCKergic upstream of PVN, AAV2/2-Retro-hEF1a-DIO-EYFP-WPRE-pA (viral titers: 2.52×10$^{13}$ particles/ml; Taitool Bioscience) virus was unilaterally injected into the PVN (AP, –0.4 mm; ML, –0.2 mm; DV, –4.85 mm) of *Cck-cre* mice.

For LPB$^{CCK}$-PVN axon terminal stimulation, AAV2/9-hEF1a-DIO-hChR2(H134R)-EYFP-WPRE-pA (viral titers: 2.95×10$^{13}$ particles/ml; Taitool Bioscience) or AAV2/9-hEF1a-DIO-EYFP-WPRE-pA (viral titers: 1.0×10$^{12}$ particles/ml; Taitool Bioscience) virus was unilaterally microinjected into the LPB (AP: –4.8; ML: –1.35; DV: –3.4; mm relative to bregma) of *Cck-cre* mice. The virus was diluted into 5.9×10$^{12}$ genomic copies per ml with PBS before use and injected with 65 nl into the LPB. After the virus was expressed for 2 weeks, for LPB$^{CCK}$-PVN axon terminals stimulation, optical fibers were unilaterally implanted above the PVN (AP: –0.4; ML: –0.2; DV: –4.7; mm relative to bregma).

For pharmacological experiments (procedures of optical activation of LPB$^{CCK}$ neurons have been mentioned above), drug cannulas were ipsilaterally implanted into the PVN (AP, –0.4 mm; ML, –0.2 mm; DV, –4.7 mm).

For prolonged inhibition of PVN neurons, AAV2/9-hSyn-mCherry-WPRE-pA (viral titers:≥1.0 × 10$^{13}$ particles/ml; Taitool Bioscience) or AAV2/9-hSyn-hM4D(Gi)-mCherry-WPRE-pA (viral titers:≥1.0 × 10$^{13}$ particles/ml; Taitool Bioscience) virus was microinjected into the PVN (AP, –0.4 mm; ML, –0.2 mm; DV, –4.85 mm) of *Cck-cre* mice.

For fiber photometry experiments, AAV2/9-hSyn-DIO-GCaMP7s-WPRE virus (viral titers: 2.0×10$^{12}$ particles/ml; Taitool Bioscience) were injected into the LPB of CCK-ires-Cre mice. After two weeks, an optical fiber was implanted into the LPB, then each mouse was allowed to recover for 1 week before recording. Each mouse was handled for 3 days prior to fiber photometry recording.

## Fiber photometry

Mice were allowed to recover from surgery for at least 7 days before the behavioral experiments. The fiber photometry system (RWD Life Science Co., Ltd, China) was used for recording fluorescence signal (GCaMP7s and isosbestic wavelengths) which produced by an exciting laser beam from 470 nm LED light and 410 nm LED light. Calcium fluorescence signals were acquired at 60 Hz with alternating pluses of 470 nm and 410 nm light. The power at the end of the optical fiber (200 μm, 0.37NA, 2 m) was adjusted to 20 μW. Recording parameters were set based on pilot studies that demonstrated the least amount of photobleaching, while allowing for the sufficient detection of the calcium response. We used the camera for behavioral video recordings to synchronize calcium recordings. On the experimental day, mice were allowed to acclimate in the home cage for 30 min. Regarding quantification, the filtered 410 nm signal was aligned with the 470 nm signal by using the least-square linear fit. ΔF/F was calculated according to (470 nm signal-fitted 410 nm signal)/(fitted 410 nm signal). And the standard z-score calculation method is used, that is, Z-score = (x-mean)/std, x = △F/F. During the behavior experiments, the GCaMP7s fluorescence intensity was recorded.

## In vivo optogenetic manipulation

For optogenetic manipulation experiments, an implanted fiber was connected to a 473 nm laser power source (Newdoon Inc, Hangzhou, China). The power of the blue (473 nm; Newdoon Inc, Hangzhou, China) was 0.83–3.33 mW mm$^2$ as measured at the tip of the fiber. 473 nm laser (ChR2: power

5–15 mW, frequency 20 Hz, pulse width 5ms; GtACR1: power 15 mW, direct current) was supplied to activate or inhibit neurons, respectively.

## Pharmacological antagonism

Antagonists were delivered 30 min before optical activation of LPB$^{CCK}$ neurons. For blocking glutamatergic neurotransmission, we firstly connected guiding cannula with a Hamilton syringe via a polyethylene tube. Then we infused 0.25 µl mixed working solution containing CNQX disodium salt hydrate (0.015 µg; Sigma-Aldrich), a glutamate AMPA receptor antagonist, and AP5 (0.03 µg; Sigma-Aldrich), a NMDA receptor antagonist, into the PVN with a manual microinfusion pump (RWD, 68606) over 5 min. For blocking CCKergic neurotransmission, 0.25 µl mixed solution containing Devazepide (0.0625 µg; Sigma-Aldrich), a *Cckar* receptor antagonist, and L-365,260 (0.0625 µg; Sigma-Aldrich), a *Cckbr* receptor antagonist, was infused into the PVN over a period of 5 min. To prevent backflow of fluid, the fluid-delivery cannula was left for 10 min after infusion.

## Chemogenetic manipulation

Clozapine Noxide (CNO, Sigma) was dissolved in saline (5 mg in 10 µL DMSO and 190 µL 0.9% NaCl solution). CNO was injected intraperitoneally at 0.3 mg per kg of body weight for chemogenetic manipulation.

## c-Fos staining and analysis

For c-fos quantification, mice were perfused 1.5 hr after 10 min blue photostimulation illumination, and sections were cut. The boundaries of the nuclei were defined according to brain atlases Mouse Brain Atlas (Franklin and Paxinos, 2008). Cell counting was carried out manually.

## Behavioral task

For all behavioral tests, experimenters were blinded to genotypes and treatments. Mice were handled daily at least seven days before performing behavioral tests. All the apparatuses and cages were sequentially wiped with 70% ethanol and ddH$_2$O then air-dried between stages. At the end of behavioral tests, mice were perfused with 4% PFA followed by post hoc analysis to confirm the viral injection sites, optic fiber and cannula locations. Mice with incorrect viral injection sites, incorrect positioning of optical fibers or cannula were excluded.

## Real-time place aversion test

Mice were habituated to a custom-made 20×30 × 40 cm two-chamber apparatus (distinct wall colors and stripe patterns) before the test. First stage: each mouse was placed in the center and allowed to explore both chambers without laser stimulation for 10 min. After exploration, the mouse indicated a small preference for one of the two chambers. Second stage: 473 nm laser stimulation (20 Hz, 8 mW, 5ms) was delivered when the mouse entered or stayed in the preferred chamber, and the light was turned off when the mouse moved to the other chamber for 10 min.

## Flight-to-nest test

Flight-to-nest test was performed using previously described methods (Zhou, Z., et al, 2019). Flight-to-nest test was performed in a 40×40 × 30 cm closed box with a 27-inch LED monitor stationed on top to display the stimulus. A nest in the shape of a 20 cm wide ×12 cm high triangular prism was in the corner of the closed Plexiglas box. There are two cameras, one from the top, one from the side (Logitech) that record the mouse's activity simultaneously. Briefly, on day 1, the mice were habituated to the box conditions for 15 min. On day 2, the mice were first allowed to explore the box for 5–10 min. When the mice were in the corner furthest from the nest and within in a body-length distance from the wall, they were given optical stimulus. For optogenetic activation of LPB$^{CCK}$ neurons or LPB$^{CCK}$-PVN experiments, mice received a 20 s 473 nm blue laser (frequency 20 Hz, pulse width 5ms) with 15–20 mW (terminal) or 5–10 mW (soma) light power at the fiber tips. For prolonged inhibition of PVN neurons, Clozapine Noxide (CNO, Sigma-Aldrich) was dissolved in saline to a concentration of 3 mg/ml. The flight-to-nest test was performed 1.5 hr after CNO intraperitoneally injection.

## Flight-to-nest behavioral analysis

The behaviors of the mice were recorded and analyzed automatically with Anymaze software (Global Biotech Inc). Behavioral analysis was performed as previously described (Zhou, Z., et al, 2019). Flight-to-nest behavior was characterized on the basis of the three aspects: latency to return nest, speed (% of baseline speed), time spent in nest (% of 2 min). Latency to return nest refers to the moment from optical stimulus onset to moment when the mouse first went into the nest. Speed (% of baseline speed) refers to the ratio of the post-stimulation speed to the baseline speed. We recorded the speed of the mice in the 50 s before photostimulation presentation as the baseline speed. The post-stimulation speed was record from the time of stimulation to 15 s after. It was averaged over a 1 s time window centered on the maximum speed. Time spent in nest (% of 2 min) refers to the time from mouse's body was first completely under the shelter after photostimulation to the time when the mouse left the nest.

## Open-field test

The open-field chamber was made of plastic (50 × 50 × 50 cm). At the start of the test, mice were placed in the periphery of the open-field chamber. The open-field test lasted 5 min.

## Elevated plus-maze test

The elevated plus maze was made of plastic with two open arms (30 × 5 cm), two closed arms (30 × 5 × 30 cm) and a central platform (5 × 5 × 5 cm). At the beginning of the experiments, mice were placed in the center platform facing a closed arm. The elevated plus-maze test lasted 5 min.

## Heart rate measurements

Heart rate was measured via a pulse oximeter (MouseOx Plus; Starr Life Sciences). Mice were placed in the home cage with a detector fixed around the neck. After habituation for 10 min, heart rate was simultaneously measured for 10 min while intermittent 2 min period blue light was applied. Each mouse was tested three times, and the mean heart rate was calculated. Heart rate data was analyzed with the MouseOx Plus Conscious Applications Software.

## Pupil size measurements

Mice were adapted to constant room light (100 lx) for 1 hr before testing. Mice were kept unanaesthetized and restrained in a stereotaxic apparatus during the experiment. The pupil size was recorded using Macro module camera under constant light conditions before stimulation, during stimulation, and after stimulation. The test lasted 90 s, consisting of 30 s light off-on-off epochs. The pupil size was later measured by Matlab software.

## Heat exposure assay

We used a translucent plastic box (38×25 cm) divided into two parts, the smaller part is the unheated comfortable zone (5×25 cm), and the larger part with rubber heating pad is the hot uncomfortable zone (33×25 cm). The rubber heating pad was heated to 43°C, and the heat insulated pad was placed in the comfortable zone to avoid the heat. The heating temperature 43°C was chosen because it would cause heat escape but not heat pain (*Wang et al., 2021b*; *Weisheng et al., 2021*).

## Rat exposure assay

We used a rectangular chamber (70×25 × 30 cm). Mice were acclimated to this environment for three days for 10 min each day. During the rat exposure period, a live rat was restrained to one end of the chamber using a harness attached to the chamber wall. As a control, before a live rat exposure, we exposed mice to a toy rat during fiber photometry recording (similar in size and shape to a live rat). For fiber photometry recordings, all mice underwent rat exposure for 20 min. For photoinhibition tests, mice were exposed to live rat and all trials lasted 10 min.

## Looming test

The looming test was performed in a closed Plexiglas box (40×40 × 30 cm) with a shelter in the corner. For looming stimulation, an LCD monitor was placed on the ceiling to present multiple looming stimuli, which was a black disc expanding from a visual angle of 2° to 20° in 0.5 s. The expanding disc stimulus

was repeated for 20 times in quick succession and each repeat is followed by a 0.15 s pause. Animals were habituated for 10–15 min in the looming box one day before testing. During the looming test, mice were first allowed to freely explore for 3–5 min. For calcium signal experiment, total five trials of looming stimuli were presented and analyzed. Behavior was recorded for 20 min. For photoinhibition tests, light stimulation was given 1 s before the looming stimulus appears and continue until the looming stimulus ends.

## TMT odor test

During this test, all mice were habituated to the testing environment for three days before any experimental manipulation. For the photometry studies involving odor presentations, mice were placed on a plastic chamber (30×30 × 20 cm). A dish (diameter, 5 cm) with cotton was positioned on the side beside the chamber. When TMT (Ferro Tec, 4 µl of 100%) was used as the stimulus, cotton was first wetted with the equivalent volume of 0.1 M PBS. After a habituation period, the dish was replaced with a new one scented with the TMT. The mice were free to explore in the chamber, and behavior was recorded for 20 min.

## Measurements of corticosterone

Blood samples were collected to determine the hormone levels. Taking Blood was performed in the morning by rapidly collecting heart blood after anesthetization. We immediately collected blood after 10 min of light stimulation ( 20 Hz, 5ms pulse width, 15 s per min, 473 nm). Blood samples were temporarily placed in iced plastic tubes coated with heparin. All serum was prepared after every blood sample was centrifuged at 2000 g for 2.5 min at 4 °C. Supernatant was collected and plasma corticosterone concentration was measured using commercially-available ELISA kits (Enzo ADI-900–097).

## In vitro electrophysiology

Each mouse was anesthetized with pentobarbital sodium (100 mg/kg, i.p.) and decapitated. Then the whole brain was quickly dissected into ice-cold oxygenated (95% $O_2$ and 5% $CO_2$) artificial cerebrospinal fluid (aCSF) (93 mM N-methyl-D-glucamine, 2.5 mM KCl, 1.2 mM $NaH_2PO_4$, 20 mM HEPES, 25 mM D-glucose, 30 mM $NaHCO_3$, 10 mM $MgSO_4$, 0.5 mM $CaCl_2$, 5 mM sodium ascorbate, 3 mM sodium pyruvate, and 1 mM kynurenic acid), followed by cutting coronally into 300 µm slices on a microtome (VTA-1200S; Leica). Slices containing the LPB were transferred to a similar solution (93 mM NaCl, 2.5 mM KCl, 1.2 mM $NaH_2PO_4$, 20 mM HEPES, 25 mM D-glucose, 30 mM $NaHCO_3$, 2 mM $MgSO_4$, 2 mM $CaCl_2$, 5 mM sodium ascorbate, 3 mM sodium pyruvate, and 1 mM kynurenic acid), and incubated for at lowest 1 h at room temperature (24–26°C). Then the brain slices were transferred to a recording chamber attached to the fixed stage of an BX51WI microscope (Olympus) (solution containing: 125 mM NaCl, 3 mM KCl, 1.25 mM $NaH_2PO_4$, HEPES, 10 mM D-glucose, 26 mM $NaHCO_3$, 2 mM $MgSO_4$ and 2 mM $CaCl_2$). Patch glass electrodes were pulled from borosilicate capillaries (BF150-86-10; Sutter Instrument Co, Novato, CA, USA) and filled with artificial intracellular fluid following component: 135 mM $CsMeSO_3$, 10 mM HEPS, 0.5 mM EGTA, 3.3 mM QX-314, 4 mM Mg-ATP, 0.3 mM $Na_2$-GTP, 8 mM $Na_2$-Phosphocreatine. Whole-cell voltage-clamp recordings were made with a Multi-Clamp 700B amplifier (Molecular Devices). To tested the efficacy of ChR2-mediated activation, LED-generated blue light pulses were applied to recorded neurons using 4 different frequencies (5, 10, 20, and 40 Hz). To test the effects of photoactivation of LPB$^{CCK}$ projection terminals within PVN, blue light pulses were applied to the recorded PVN neurons. To confirm that postsynaptic currents were monosynaptic, the blue light-evoked currents were recorded in the presence of ACSF (Ctrl), TTX (1 µM) and 4-AP (100 µM). To confirm that postsynaptic currents were monosynaptic. CNQX (20 µM) and AP5 (50 µM) were perfused with ACSF to examine the neurotransmitter type used on LPB$^{CCK}$-PVN projection. Signals were low-pass filtered at 10 kHz and digitized at 10 kHz (MICRO3 1401, Cambridge Electronic Design). Data were acquired and analyzed using Spike2 7.04 software (Cambridge Electronic Design).

## Quantification and statistical analysis

All data analyses were conducted blinded. All statistical analyses were performed with GraphPad Prism (version 7.0) and analyzed by Unpaired Student's t-tests, one-way ANOVA, two-way ANOVA according to the form of the data. Nonparametric tests were used if the data did not match assumed

Gaussian distribution. Animals were randomly assigned to treatment groups. All data were presented as Mean ± SEM, with statistical significance taken as *p<0.05, **p<0.01 and ***p<0.001.

## Acknowledgements

We thank Xiaoming Li, Weisheng Wang, Bin Zhang, Chenjie Shen, Zhi Chen, Yu-dong Zhou, Kai Liu, Huifang Lou for technical support. This work was supported by STI2030-Major Projects (2021ZD0202700 and 2021ZD0202703), the Leading talents in Science and Technology of Zhejiang Province (2021R52021), the National Natural Science Foundation of China (82090033), the NSFC-Guangdong Joint Fund (U20A6005), the Zhejiang Major Science and Technology Project (2020C03009 and 2020R01001), the Research Units for Emotion and Emotion Disorders, Chinese Acedemy of Medical Sciences (2019-I2M-5–057), the Key Area Research and Development Program of Guangdong Povince (2019B030335001).

## Additional information

### Funding

| Funder | Grant reference number | Author |
|---|---|---|
| STI2030-Major Projects | 2021ZD0202700 | Zhihua Gao |
| STI2030-Major Projects | 2021ZD0202703 | Zhihua Gao |
| Leading Talents in Science and Technology of Zhejiang Province | 2021R52021 | Zhihua Gao |
| National Nature Science Foundation of China | 82090033 | Shumin Duan |
| NSFC-Guangdong Joint Fund | U20A6005 | Shumin Duan |
| Zhejiang Major Science and Technology Project | 2020C03009 | Shumin Duan |
| Zhejiang Major Science and Technology Project | 2020R01001 | Shumin Duan |
| Chinese Academy of Medical Sciences | Research Units for Emotion and Emotion Disorders, 2019-I2M-5–057 | Shumin Duan |
| Key Area Research and Development Program of Guangdong Province | 2019B030335001 | Shumin Duan |

The funders had no role in study design, data collection and interpretation, or the decision to submit the work for publication.

### Author contributions

Fan Wang, Yuge Chen, Conceptualization, Resources, Data curation, Software, Formal analysis, Supervision, Validation, Investigation, Visualization, Methodology, Writing – original draft, Project administration, Writing – review and editing; Yuxin Lin, Conceptualization, Resources, Data curation, Formal analysis, Methodology, Writing – review and editing; Xuze Wang, Conceptualization, Resources, Data curation, Software, Formal analysis, Investigation, Methodology, Project administration, Writing – review and editing; Kaiyuan Li, Jintao Wu, Data curation, Software; Yong Han, Zhenggang Zhu, Xiaojun Hu, Visualization, Writing – original draft; Xingyi Shi, Data curation, Software, Formal analysis; Chaoying Long, Data curation; Shumin Duan, Zhihua Gao, Conceptualization, Resources, Data curation, Formal analysis, Supervision, Funding acquisition, Validation, Investigation, Visualization, Methodology, Writing – original draft, Project administration, Writing – review and editing

### Author ORCIDs

Zhihua Gao http://orcid.org/0000-0001-8603-0777

## Ethics

Animal experimentation: All experimental procedures were approved by the Animal Advisory Committee at Zhejiang University and were performed in strict accordance with the National Institutes of Health Guidelines for the Care and Use of Laboratory Animals. Laboratory Animal Welfare and Ethics Committee of Zhejiang University; National Institutes of Health Guidelines for the Care and Use of Laboratory Animals; AP CODE:ZJU20220160.

## Decision letter and Author response

Decision letter https://doi.org/10.7554/eLife.85450.sa1
Author response https://doi.org/10.7554/eLife.85450.sa2

---

# Additional files

## Supplementary files

• MDAR checklist

## Data availability

All data generated or analysed during this study are included in the manuscript and supporting file. Source data files have been uploaed.

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
