## [Editor Report]

In this study, the authors revealed that activation of LPB CCK-expressing neurons could drive flight-to-nest behavior and increase sympathetic output while inhibiting the cells reduces the behavioral response to predator exposure and visual predatory cues. They further found that LPB CCK neurons project to the PVN area, and activation of this pathway caused similar behavioral changes. Lastly, activating PVN glutamatergic cells can also induce flight. The evidence is solid, the study is valuable, and it will be of interest to the affective and circuit neuroscience fields as it provides insights into a novel parabrachial to hypothalamus pathway that could potentially mediate threat avoidance behavior.

---

## [Decision Letter]

**Decision letter after peer review:**

[Editors’ note: the authors submitted for reconsideration following the decision after peer review. What follows is the decision letter after the first round of review.]

Thank you for submitting the paper "A Parabrachial to Hypothalamic Neurocircuit Mediates Defensive Behavior" for consideration by *eLife*. Your article has been reviewed by 3 peer reviewers, and the evaluation has been overseen by a Reviewing Editor and a Senior Editor. The reviewers have opted to remain anonymous.

Comments to the Authors:

We are sorry to say that, after consultation with the reviewers, we have decided that this work will not be considered further for publication by *eLife*.

Specifically, all reviewers feel that the current study did not convincingly reveal the endogenous function of the LPBCCK to PVHglut pathway. In addition, the major conclusions are not fully supported by the experimental data.

However, if you can fully address the reviewer's concerns, we would be happy to encourage you to submit a revised manuscript as a new submission. We hope that you will find the reviewers' comments to be constructive and if you have any questions about the reviews please let us know.

*Reviewer #1 (Recommendations for the authors):*

The behavioral results of optogenetic activation are overall robust. However, the role of the cells in the defense circuit remains unclear due to a lack of information regarding the endogenous responses of the cells. Furthermore, while the projection from LPB cck to PVN is nicely established, the manipulation at PVN is less convincing due to the broad expression of vglut2, and thus cre, in the regions surrounding PVN.

1. Natural responses of cells are completely missing in the study. Do PB CCK cells or non-CRH PVN glutamatergic cells increase activity during stressful conditions? Such as noxious heat? It is unclear what is the role of the PB cck cells in the defense circuit. Is it primarily a sensory region for aversive stimulus or a motor region for driving the behavioral output?

2. The behavior paradigm used for the loss-of-function experiment is somewhat unusual. A looming stimulus is more commonly used to elicit flight in mice. Does inhibiting PB cck cells also suppress looming induced flight? This result could be interesting either way. It will help elucidate the natural role of the cells in defensive behavior.

3. For Figure 1, please show the histology of the primary injection site and quantify the results. Given that AAV2-retro is not cre-dependent and PVN is a small region, it is not clear how many of the retrogradely labeled cells are due to their projection to the PVN vs. PVN surrounding region. For figure 1c, is this number from the sum of 3 animals or per animal? If it is per animal, what is the level of variability across animals?

4. For Figure 2I, please show the quantification of TTX+4AP results.

5. Does terminal activation at PVN also induce autonomic responses? The answer to this question helps to address whether the autonomic and behavioral changes are mediated by the same or distinct pathways.

6. Optogenetic experiments allow for within-animal comparison. Please analyze the sham light period for ChR2 and GFP groups.

7. Based on Figure 8B histology, there is quite a bit of infection outside of PVN. Please quantify the extent of the virus spread. Given that Vglut2 is not a PVN-specific marker, this raises the concern regarding whether the behavioral change is indeed due to the activation of PVN cells. If PB cck cells do not target onto PVN CRH cells, what do they target onto? Oxy cells? AVP cells? Or something else? The authors need to provide more convincing evidence that the behavior change is due to the activation of PVN glutamatergic cells.

*Reviewer #2 (Recommendations for the authors):*

This manuscript described a neuronal pathway from CCK expressing neurons in the lateral parabrachial nucleus (LPBCCK) to glutamatergic neurons in the paraventricular hypothalamus (PVHglut) for defensive behaviors. The authors provided convincing data showing that artificially activating this pathway can drive defensive-like flight to nest behavior and can increase anxiety and drive autonomic responses in mice. However, how this pathway engaged in physiological defensive responses is not known. The experiment for examining the role of LPBCCK neurons in heat defensive behavior has flaws. LPBCCK neurons itself is important for thermoregulation. Therefore, the increased latency for the mice to avoid heat area after silencing LPBCCK neurons could simply be because the mice become less sensitive to heat, but not necessarily lack defensive behavior. Also because Yang et al., have already reported the role of the LPBCCK neurons in heat regulation, the novelty of the discovery in the current manuscript is questioned. The flight to nest behavior is likely to be also triggered by looming stimuli. Does the LPBCCK to PVHglut pathway also responsible for flighting responses triggered by the looming stimuli?

My biggest concern for this manuscript is the lacking of evidence to support an endogenous function of the LPBCCK to PVHglut pathway.

Also, more characterization of both LPBCCK and PVHglut neurons is needed. Did these neurons also express other markers that are well known in these two regions?

It is surprising to me that no in vivo recording has been done on this pathway in defensive behavior.*Reviewer #3 (Recommendations for the authors):*

The current manuscript will be of high interest to the affective and circuit neuroscience fields as it provides insights into a novel parabrachial to hypothalamus pathway that could potentially mediate threat avoidance behavior. This study used comprehensive and clever approaches to attempt to delineate the involvement of this pathway using a rodent task that assesses escape and shelter-seeking behavior. However, the authors still need to rule out alternative explanations to determine with more certainty that the behavior mice display is indeed threat avoidance and safety-seeking.

The major strengths of this manuscript are that the authors did a very nice and thorough job demonstrating LPBCCK – PVN pathway connectivity. The authors also did a very good job showing the sufficiency of this pathway in producing escape-like behavior in mice.

The main concern I have with this manuscript is that as it is, and with the behavioral paradigms used, the authors cannot fully claim that this LPBCCK – PVN pathway plays a role in defensive-like avoidance behavior as alternative explanations first need to be ruled out before making this claim.

As the authors noted throughout the manuscript, previous studies have shown that LPBCCK neurons project to the VMH and mediate the counterregulatory responses produced by extreme hypoglycemia. Such hypoglycemia-induced counterregulatory responses result in the secretion of several hormones, including CORT (Garfield et al., 2014; Ritter et al., 2019; Beas et al., 2020). These counterregulatory responses are also very stressful and produce anxiety-like effects in rodents (Ritter et al., 2019; Beas et al., 2020). As such, it is very plausible that the flight-to-nest behavior the authors observed after activation of the LPBCCK – PVN pathway could have similar aversive effects that result in escape behavior but that are unrelated to defensive avoidance.

My second concern pertains to the thermoregulation paradigm used to test the requirement of LPBCCK neurons on avoidance behavior (as shown in Figure 5). Using thermos-sensation as a paradigm to assess avoidance behavior is a clever way to test the requirement of these LPBCKK neurons. However, equating heat avoidance to defensive-like treat avoidance behavior might be problematic since predator escaping and thermo-regulation are known to be mediated by different pathways (Lek Tan et al., 2018; Yang et al., 2020).

Overall, this manuscript could undoubtedly benefit from differentiating whether the escape-avoiding behavior displayed by manipulating the LPBCCK – PVN pathway results from perceived interoceptive stressors (hypoglycemia, changes in temperatures, etc.), exteroceptive threats (predator smell, predator looming, etc.), or both. Doing so could provide more insight into the actual functional role of this LPBCCK – PVN pathway and would likely have a more substantial impact on the field.

To be able to rule out the involvement of this pathway in mediating the stressful responses of perceived hypoglycemia, authors could show that their LPBCCK – PVN manipulations target a different subset of VGlut2 neurons in the PVN from those expressing SF-1 (Garfield et al., 2014). Authors could also use similar approaches as in Garfield et al., 2014, such as 2DG injections or insulin as models of hypoglycemia while inhibiting the LPBCCK – PVN pathway. Suppose their manipulations do not affect measurements such as food consumption, CORT, and glucose levels after administering 2DG or insulin, as in Garfield et al., 2014; the authors can then conclude that the activation of this pathway drives defensive-like avoidance behavior, and not an aversive state produced by a perceived hypoglycemic state.

In addition, if the authors hypothesize that the LPBCCK – PVN pathway plays a role in defensive-like avoidance behavior and are aiming to test the requirement of these neurons on threat avoidance behavior, they could use predator odor or looming objects such as those used in Zhou et al., 2019 which are known to produce escape behaviors. Using these paradigms would allow them to better examine the effects of optogenetic inhibition of LPBCCK neurons on threat avoidance/shelter-seeking behavior.

Lastly, in figures 5F and 5G, there seems to be a bimodal distribution of the mice for the GtACR1 group. One group of mice shows a profound impairment in thermal seeking behavior after inhibition, while another group only shows a slight effect. Could optical fiber location within the LPBN be accounting for these differences? As such, it would be necessary for the authors to show fiber implant locations for all experiments.

[Editors’ note: further revisions were suggested prior to acceptance, as described below.]

Thank you for resubmitting your work entitled "A Parabrachial to Hypothalamic Pathway Mediates Defensive Behavior" for further consideration by *eLife*. Your revised article has been evaluated by Kate Wassum (Senior Editor) and a Reviewing Editor.

The manuscript has been improved but there are some remaining issues that need to be addressed, as outlined below:

Please address the important points noted by Reviewer #1 below. Maps of fiber placements and viral expression should be included in all relevant experiments.

Please ensure your manuscript complies with the *eLife* policies for statistical reporting: https://reviewer.elifesciences.org/author-guide/full "Report exact p-values wherever possible alongside the summary statistics and 95% confidence intervals. These should be reported for all key questions and not only when the p-value is less than 0.05." in particular, statistics were not readily identified for Figure 1P, 3, N, O, Q, R.

*Reviewer #1 (Recommendations for the authors):*

The authors addressed most of my concerns.

However, I still have a couple of minor concerns/recommendations:

1. Figure 4 D-H – During the rat exposure behavioral test, the authors show that optogenetic inhibition of LPBCCK neurons results in increases in entries and time in the 'danger zone.' However, are there any changes in latencies to flight and flight speed after silencing these neurons?

2. For rigor purposes, the author should show anatomical maps displaying fiber placements for all the optogenetics and fiber photometry experiments.

3. Figure 3F is missing the y-axis.

*Reviewer #2 (Recommendations for the authors):*

I appreciate the author added new experiments to address my concerns. The revised manuscript is easy to read and is ready for publication.

*Reviewer #3 (Recommendations for the authors):*

The authors did a good job addressing the reviewer's comments. The paper has been improved significantly.

---

## [Author Response]

Comments to the Authors:Reviewer #1 (Recommendations for the authors):The behavioral results of optogenetic activation are overall robust. However, the role of the cells in the defense circuit remains unclear due to a lack of information regarding the endogenous responses of the cells. Furthermore, while the projection from LPB cck to PVN is nicely established, the manipulation at PVN is less convincing due to the broad expression of vglut2, and thus cre, in the regions surrounding PVN.1. Natural responses of cells are completely missing in the study. Do PB CCK cells or non-CRH PVN glutamatergic cells increase activity during stressful conditions? Such as noxious heat? It is unclear what is the role of the PB cck cells in the defense circuit. Is it primarily a sensory region for aversive stimulus or a motor region for driving the behavioral output?

We very much appreciate the reviewer’s questions.

1) This was indeed a big disadvantage of our study and reviewer #2 also raised this concern (Reviewer #2, point 1). We have fixed this problem in the updated manuscript by recording calcium responses of LPB^CCK^ neurons in vivo under different conditions, including heat, predatory (rat or TMT) exposure and looming stimuli. In accordance with previous studies (Yang et al., 2020), we verified that LPB^CCK^ neurons were activated when mice were exposed to heat (Revised Supplementary Figure 4A-D). Importantly, rat exposure, predatory cues TMT and looming stimuli also robustly activated LPB^CCK^ neurons (Revised Figure 3 A-R and Supplementary Figure 4I-L). Of note, the calcium signals of LPB^CCK^ neurons started to rise when mice assessed the threat, reached the peak at the moment of flight initiation, but rapidly dropped when mice gained distance from the threat.

2) We performed c-fos staining after stimulation of the terminals of LPB^CCK^ neurons in the PVN. We found that PVN^CRH^ neurons are activated, whereas non-CRH neurons, including AVP or OXT neurons in the PVN, are not activated (Revised Supplementary Figure 5C-F).

3) We also performed the real-time place aversion (RTPA) tests and found that activation of PB^CCK^ neurons induces apparent aversion (Revised Figure 2E-G). Given that LPB is an important sensory hub and LPB^CCK^ neurons are recruited to different threat stimuli, our data support that L

4) LPB^CCK^ neurons primarily act as a sensory and transmitting region for aversive stimuli (This has been further elaborated in the Discussion section, Line 348-353).

2. The behavior paradigm used for the loss-of-function experiment is somewhat unusual. A looming stimulus is more commonly used to elicit flight in mice. Does inhibiting PB cck cells also suppress looming induced flight? This result could be interesting either way. It will help elucidate the natural role of the cells in defensive behavior.

We agree with the reviewer that the heat avoidance paradigm may not be optimal in our original study. We have removed the heat avoidance assay in the revised manuscript. We have used both looming stimuli and rat predatory tests to examine the role of LPB^CCK^ neurons in flight responses. As shown in the Revised Figure 4D-L, optogenetic inhibition of LPB^CCK^ neurons significantly attenuated looming stimuli or rat exposure-triggered flight responses.

3. For Figure 1, please show the histology of the primary injection site and quantify the results. Given that AAV2-retro is not cre-dependent and PVN is a small region, it is not clear how many of the retrogradely labeled cells are due to their projection to the PVN vs. PVN surrounding region. For figure 1c, is this number from the sum of 3 animals or per animal? If it is per animal, what is the level of variability across animals?

We thank the reviewer for this question.

1) The virus we injected into the CCK-Cre mice is Cre-dependent AAV2-retro-DIO-EYFP.

2) Since the fluorescence of AAV2-retro-DIO-EYFP is barely invisible in the PVN in CCK-Cre mice after injection, we often co-injected the virus with the retrograde tracer CTB to mark the infected area. The data showing the injection site and CTB-labeled area in the PVN have now been included in the Revised Figure 1B.

3) We usually titrate the amount of injected virus to find out an optimal does for the retrograde tracing (Zhang et al., 2021).Due to the invisibility of fluorescence signals of AAV2-retro-DIO-EYFP, however, we can only judge the accuracy of injection and infected area based on the CTB^+^ area within the PVN. To minimize potential overfill out of the PVN regions after virus infection, only animals showing CTB^+^ signals within the PVN were included in our analyses.

4) The number shown in Figure 1c represents the average number of cells obtained from 3 animals. Cells were counted from at least three slices in individual EYFP^+^ brain regions. As shown in Author response table 1, there are variations in different animals, but the percentages of traced cells are quite consistent across animals (Revised Figure 1D).

**Author response table 1. sa2table1:** Numbers of retrogradely traced cells in CCK-Cre mice.

Animal 1						
Region	Slice 1	Slice 2	Slice 3	total	percentage	average
CC	94	103	95	292	25.75%	97
DI	11	34	21	66	5.82%	22
LPB	71	103	69	243	21.43%	81
PAG	26	34	19	79	6.97%	26
MO	137	175	142	454	10.04%	151
Animal 2						
Region	Slice 1	Slice 2	Slice 3	total	percentage	average
CC	82	93	75	250	21.82%	83
DI	18	23	17	58	5.06%	19
LPB	94	102	91	287	25.04%	96
PAG	11	23	15	49	4.28%	16
MO	166	172	164	502	43.80%	167
Animal 3						
Region	Slice 1	Slice 2	Slice 3	Total	percentage	average
CC	80	95	76	251	24.25%	84
DI	27	32	28	87	8.41%	29
LPB	67	82	63	212	20.48%	71
PAG	12	19	16	47	4.54%	16
MO	139	157	142	438	42.32%	146
Region	Total average					
CC	88					
DI	23					
LPB	83					
PAG	19					
MO	155					

MO, medial orbital cortex; CC, cingulate cortex; DI, dysgranular insular cortex; PAG, periaqueductal gray

4. For Figure 2I, please show the quantification of TTX+4AP results.

We have included the quantification of TTX+4AP results in Revised Figure 1N. Please note, due to the changes of figure orders, the quantification data is now presented in Figure 1N, instead of the original Figure 2I.

5. Does terminal activation at PVN also induce autonomic responses? The answer to this question helps to address whether the autonomic and behavioral changes are mediated by the same or distinct pathways.

Yes, terminal activation at the PVN of LPB^CCK^ neurons induced autonomic responses as it increased heart rates and plasma corticosterone levels (Revised Figure 5K-N). However, terminal stimulation barely affected pupil size. These data suggest that different aspects of autonomic responses are likely mediated by different pathways downstream of LPB^CCK^ neurons. We appreciate the reviewer’s point, and we are further investigating whether different downstream pathways of LPB^CCK^ neurons mediate different aspects (autonomic and behavioral components) of the complex defensive responses.

6. Optogenetic experiments allow for within-animal comparison. Please analyze the sham light period for ChR2 and GFP groups.

We thank the reviewer for raising this important point. We have carefully compared the animal behaviors before and during/after light stimulation for both ChR2 and EYFP groups in the real-time place aversion (RTPA), anxiety-related tests. These data are now presented in the Revised Figure 2G, Supplementary Figure 3D-J. These data nicely present the behavioral responses of animals during/after light stimulation of LPB^CCK^ neurons. For the flight-to-nest paradigm, however, since there is no flight latency and nest-residing time before light stimulation, we kept the original analyses between EYFP and ChR2 groups.

7. Based on Figure 8B histology, there is quite a bit of infection outside of PVN. Please quantify the extent of the virus spread. Given that Vglut2 is not a PVN-specific marker, this raises the concern regarding whether the behavioral change is indeed due to the activation of PVN cells. If PB cck cells do not target onto PVN CRH cells, what do they target onto? Oxy cells? AVP cells? Or something else? The authors need to provide more convincing evidence that the behavior change is due to the activation of PVN glutamatergic cells.

We thank the reviewer for raising this concern.

1) We agree with the reviewer that vGlut2 is not a PVN-specific marker, as it is widely distributed in all glutamatergic neurons. Studies have shown that, however, PVN neurons are predominantly vGluT2^+^ cells (Vong et al., 2011). In addition, ISH data from the Allen Brain Atlas (https://portal.brain-map.org/) also showed that Vglut2 neurons are located in the PVN, whereas GABAergic neurons are almost excluded from the PVN region by surrounding the PVN (Author response image 1). By expressing ChR2 in PVN^Vglut2^ neurons and putting optic fibers above this area, we believe that we primarily activated cells in the PVN. In support of our data, activation of Sim1^+^ (a specific marker for PVN) cells in the PVN also promoted flight behaviors(Mangieri et al., 2019), similar to the behavioral responses we observed after activation of PVN^Vglut2^ neurons.

**Author response image 1. sa2fig1:** ISH data from the Allen Brain Atlas.

2) To find out which type of cells in the PVN receives input from LPB^CCK^ neurons, we further carried out c-fos staining in PVN regions after stimulation of LPB^CCK^-PVN terminals. Our data demonstrate that PVN^CRH^ neurons, but not AVP, OXT neurons are activated, suggesting that LPB^CCK^ neurons do target onto PVN^CRH^ cells. While activation of PVN^CRH^ cells have been shown to promote aversion (Kim et al., 2019), we were unable to drive apparent flight-to-nest responses in our assay. We are further investigating the potential mechanisms involved.

Reviewer #2 (Recommendations for the authors):This manuscript described a neuronal pathway from CCK expressing neurons in the lateral parabrachial nucleus (LPBCCK) to glutamatergic neurons in the paraventricular hypothalamus (PVHglut) for defensive behaviors. The authors provided convincing data showing that artificially activating this pathway can drive defensive-like flight to nest behavior and can increase anxiety and drive autonomic responses in mice. However, how this pathway engaged in physiological defensive responses is not known. The experiment for examining the role of LPBCCK neurons in heat defensive behavior has flaws. LPBCCK neurons itself is important for thermoregulation. Therefore, the increased latency for the mice to avoid heat area after silencing LPBCCK neurons could simply be because the mice become less sensitive to heat, but not necessarily lack defensive behavior. Also because Yang et al., have already reported the role of the LPBCCK neurons in heat regulation, the novelty of the discovery in the current manuscript is questioned. The flight to nest behavior is likely to be also triggered by looming stimuli. Does the LPBCCK to PVHglut pathway also responsible for flighting responses triggered by the looming stimuli?My biggest concern for this manuscript is the lacking of evidence to support an endogenous function of the LPBCCK to PVHglut pathway.

We appreciate this important question from the reviewer. This problem has now been solved by in vivo recording the calcium activity of LPB^CCK^ neurons under different conditions including heat, rat exposure, looming stimuli, and predatory cues (Please see also our response to Reviewer #1, point 1 and Revised Figure 3). Our data show that PB^CCK^ neurons respond to various innate threat stimuli.

In addition, we also tried to record the terminal calcium responses of PB^CCK^ neurons in PVN region, by injecting the AAV-DIO-synapse-jGCaMP7b virus which has been to preferentially target to axon terminals (Yang et al., 2020). However, no fluorescence signals were seen in different downstream terminals of PB^CCK^ neurons, including the PVN and parasubthalamic nucleus (PSTh) (Author response image 2). Fiber photometry recording also failed to detect any calcium responses under threat stimulation. We suspect that there might be a problem with the virus itself and we are also trying to get viruses from abroad. However, due to COVID-19 pandemic, we still haven’t received the viruses after several months’ wait. Since there are competitions regarding the role of LPB^CCK^ neurons in defensive behaviors, we have to wrap our study and submit the manuscript without further wait. We sincerely apologize for being unable to finish this experiment.

**Author response image 2. sa2fig2:** Representative figures of AAV-DIO-synapse-jGCaMP7b infection at the LPB and downstream PVN and PSTh.

Also, more characterization of both LPBCCK and PVHglut neurons is needed. Did these neurons also express other markers that are well known in these two regions?

We fully agree with the review. We are actively trying different methods, including single cell transcriptomics to fully characterize the identity of these cells. According to Shen et al. (Yang et al., 2020), LPB^CCK^ neurons participate in thermoregulation by projecting to the preoptic area (POA), whereas Norris et al. (Norris, Shaker, Cone, Ndiokho, & Bruchas, 2021) also showed that LPB neurons expressing *dynorphin* and *enkephalin* regulate body temperature by projecting to POA*.* It is likely that neuronal populations in the above studies may overlap, and LPB^CCK^ neurons may also contain *dynorphin* and *enkephalin*. Notably, although CGRP neurons have been shown to encode threat signals, our data show that LPB^CCK^ neurons barely contain CGRP (Revised Supplementary Figure 2F).

Regarding the PVH (PVN)^Vglut2^ neurons, this category likely includes the majority of PVN neurons (please see response to Reviewer 1#, point 7). PVN neurons contain a large repertoire of neuropeptides (Zhang et al., 2021), and we found that PVN^CRH^ neurons are responsive to LPB^CCK^ neurons stimulation (Supplementary Figure 5C).

It is surprising to me that no in vivo recording has been done on this pathway in defensive behavior.

We thank the reviewer for this important question. Indeed, this was a big disadvantage in our original manuscript, we have now included the in vivo recording data in the updated manuscript (Revised Figure 3A-R and Supplementary Figure 4I-L). Please see also our response to Reviewer #1, point 1.

Reviewer #3 (Recommendations for the authors):The current manuscript will be of high interest to the affective and circuit neuroscience fields as it provides insights into a novel parabrachial to hypothalamus pathway that could potentially mediate threat avoidance behavior. This study used comprehensive and clever approaches to attempt to delineate the involvement of this pathway using a rodent task that assesses escape and shelter-seeking behavior. However, the authors still need to rule out alternative explanations to determine with more certainty that the behavior mice display is indeed threat avoidance and safety-seeking.The major strengths of this manuscript are that the authors did a very nice and thorough job demonstrating LPBCCK – PVN pathway connectivity. The authors also did a very good job showing the sufficiency of this pathway in producing escape-like behavior in mice.The main concern I have with this manuscript is that as it is, and with the behavioral paradigms used, the authors cannot fully claim that this LPBCCK – PVN pathway plays a role in defensive-like avoidance behavior as alternative explanations first need to be ruled out before making this claim.As the authors noted throughout the manuscript, previous studies have shown that LPBCCK neurons project to the VMH and mediate the counterregulatory responses produced by extreme hypoglycemia. Such hypoglycemia-induced counterregulatory responses result in the secretion of several hormones, including CORT (Garfield et al., 2014; Ritter et al., 2019; Beas et al., 2020). These counterregulatory responses are also very stressful and produce anxiety-like effects in rodents (Ritter et al., 2019; Beas et al., 2020). As such, it is very plausible that the flight-to-nest behavior the authors observed after activation of the LPBCCK – PVN pathway could have similar aversive effects that result in escape behavior but that are unrelated to defensive avoidance.My second concern pertains to the thermoregulation paradigm used to test the requirement of LPBCCK neurons on avoidance behavior (as shown in Figure 5). Using thermos-sensation as a paradigm to assess avoidance behavior is a clever way to test the requirement of these LPBCKK neurons. However, equating heat avoidance to defensive-like treat avoidance behavior might be problematic since predator escaping and thermo-regulation are known to be mediated by different pathways (Lek Tan et al., 2018; Yang et al., 2020).Overall, this manuscript could undoubtedly benefit from differentiating whether the escape-avoiding behavior displayed by manipulating the LPBCCK – PVN pathway results from perceived interoceptive stressors (hypoglycemia, changes in temperatures, etc.), exteroceptive threats (predator smell, predator looming, etc.), or both. Doing so could provide more insight into the actual functional role of this LPBCCK – PVN pathway and would likely have a more substantial impact on the field.To be able to rule out the involvement of this pathway in mediating the stressful responses of perceived hypoglycemia, authors could show that their LPBCCK – PVN manipulations target a different subset of VGlut2 neurons in the PVN from those expressing SF-1 (Garfield et al., 2014). Authors could also use similar approaches as in Garfield et al., 2014, such as 2DG injections or insulin as models of hypoglycemia while inhibiting the LPBCCK – PVN pathway. Suppose their manipulations do not affect measurements such as food consumption, CORT, and glucose levels after administering 2DG or insulin, as in Garfield et al., 2014; the authors can then conclude that the activation of this pathway drives defensive-like avoidance behavior, and not an aversive state produced by a perceived hypoglycemic state.

We thank the reviewer’ for raising s suggestions. LPB^CCK^ neurons had been shown to mediate the counter-regulatory responses under extreme hypoglycemic conditions via projection to VMH. We agree with the reviewer that these counterregulatory responses to hypoglycemia also produce anxiety-like states, similar to what we observed after prolonged stimulation of the LPB^CCK^ neurons. However, we believe that acute stimulation of the LPB^CCK^-PVN pathway-induced flight-to-nest behavior represent defensive behaviors, which are distinct from the responses after activation of LPB^CCK^-VMH in that:

1) By in vivo recording of the calcium responses of LPB^CCK^ neurons, we demonstrated that LPB^CCK^ neurons are actively recruited under different threat conditions, such as rat exposure, looming stimuli and predatory cues. More importantly, the peak activity of LPB^CCK^ neurons corresponds to the onset of flight behavior, clearly demonstrating that LPB^CCK^ neurons encode flight responses.

2) On a temporal scale, defensive behaviors and regulation of blood glucose levels have distinct time dynamics. For example, it usually takes a few minutes to upregulate/downregulate blood glucose levels (Huang et al., 2022; Meek et al., 2016), however, it only takes a few seconds to mount defensive responses. In Garfield’s study, they have used chemogenetic tools to activate LPB^CCK^ neurons and observed upregulated blood glucose levels around 15-20 min (Figure 2C in Garfield’s et al. 2014), which peaks at around 1hr. In our experiments, seconds of acute light stimulation induces rapid flight or aversion response within seconds. Thus, we think the rapid flight-to-nest behaviors after stimulation of LPB^CCK^-PVN are unlikely a consequence of counterregulatory response to hypoglycemia. However, the reviewer’s point is very helpful for us to further investigate the potential role of the LPB^CCK^-VMH neurons in coordinating complex defensive responses.

In addition, if the authors hypothesize that the LPBCCK – PVN pathway plays a role in defensive-like avoidance behavior and are aiming to test the requirement of these neurons on threat avoidance behavior, they could use predator odor or looming objects such as those used in Zhou et al., 2019 which are known to produce escape behaviors. Using these paradigms would allow them to better examine the effects of optogenetic inhibition of LPBCCK neurons on threat avoidance/shelter-seeking behavior.

We are very thankful to the reviewer for these wonderful suggestions. We have used both predatory (rat or TMT exposure) and looming stimuli to test whether the LPB^CCK^ neurons regulate defensive behaviors. These classical paradigms allowed us to better characterize the role of LPB^CCK^ neurons in defensive responses. We found that activation of the LPB^CCK^-PVN pathway promotes flight-to-nest behavior, and inhibition of LPB^CCK^ neurons suppressed rat exposure or looming stimuli-triggered defensive behaviors. These data are now included in Revised Figure 4 D-L. Our new findings demonstrate that the LPB^CCK^-PVN pathway is important for the defensive responses.

Lastly, in figures 5F and 5G, there seems to be a bimodal distribution of the mice for the GtACR1 group. One group of mice shows a profound impairment in thermal seeking behavior after inhibition, while another group only shows a slight effect. Could optical fiber location within the LPBN be accounting for these differences? As such, it would be necessary for the authors to show fiber implant locations for all experiments.

We thank the reviewer for the careful observation and excellent suggestions. Indeed, after a careful re-examination of optical fiber locations in the LPB, we noticed that 3 animals showed accurate positions of optic fibers in only one side of the LPB (Author response image 3). These animals were exactly those showing no apparent avoidance behaviors in the heat-defense behavioral tests, likely due to the insufficient inhibition. As per Reviewer 2’s concerns, these data are now removed. According to all reviewers’ suggestions, we have used rat exposure or looming stimuli assays to determine the role of LPB^CCK^ neurons in defensive behaviors and these data are presented in Revised Figure 4D-L.

**Author response image 3. sa2fig3:** Representative images of fibers implantation above the LPB after injection of AAV-DIO-GtACR1-EYFP virus.

**Author response table 2. sa2table2:** Data of behavioral tests after inhibition of LPB^CCK^ neurons.

	Animal	Stage	Hot zone : time	Safe zone : latency to first entry	Hot zone time/total time	Optical fiber location
GtACR1	1	Habitat stage	96.5			Optical fibers in only one side
		Test stage	36.7	13.5	12.23%	
	2	Habitat stage	60.8			Optical fibers in only one side
		Test stage	70	67.0	23.33%	
	3	Habitat stage	13.4			
		Test stage	265.7	265.7	88.57%	
	4	Habitat stage	151.2			
		Test stage	300	300.0	100.00%	
	5	Habitat stage	176.7			
		Test stage	300	300.0	100.00%	
	6	Habitat stage	70.6			Optical fibers in only one side
		Test stage	93	91.7	31.00%	
	7	Habitat stage	33.7			
		Test stage	137.9	106.7	45.97%	
EYFP	8	Habitat stage	84.8			
		Test stage	72.9	2.3	24.30%	
	9	Habitat stage	41.9			
		Test stage	18.4	5.6	6.13%	
	10	Habitat stage	14.9			
		Test stage	6.7	2.4	2.23%	
	11	Habitat stage	77.6			
		Test stage	44.9	9.0	14.97%	
	12	Habitat stage	137.3			
		Test stage	52	7.5	17.33%	
	13	Habitat stage	75.8			
		Test stage	41.3	38.0	13.77%	
	14	Habitat stage	149.4			
		Test stage	108.9	108.9	36.30%	
	15	Habitat stage	30.8			
		Test stage	6.8	4.1	2.27%	

References

Huang, Z., Liu, L., Zhang, J., Conde, K., Phansalkar, J., Li, Z.,… Liu, J. (2022). Glucose-sensing glucagon-like peptide-1 receptor neurons in the dorsomedial hypothalamus regulate glucose metabolism. *Sci Adv, 8*(23), eabn5345. doi:10.1126/sciadv.abn5345

Kim, J., Lee, S., Fang, Y. Y., Shin, A., Park, S., Hashikawa, K.,… Suh, G. S. B. (2019). Rapid, biphasic CRF neuronal responses encode positive and negative valence. *Nat Neurosci, 22*(4), 576-585. doi:10.1038/s41593-019-0342-2

Mangieri, L. R., Jiang, Z., Lu, Y., Xu, Y., Cassidy, R. M., Justice, N.,… Tong, Q. (2019). Defensive Behaviors Driven by a Hypothalamic-Ventral Midbrain Circuit. *eNeuro, 6*(4). doi:10.1523/ENEURO.0156-19.2019

Meek, T. H., Nelson, J. T., Matsen, M. E., Dorfman, M. D., Guyenet, S. J., Damian, V.,… Morton, G. J. (2016). Functional identification of a neurocircuit regulating blood glucose. *Proc Natl Acad Sci U S A, 113*(14), E2073-2082. doi:10.1073/pnas.1521160113

Norris, A. J., Shaker, J. R., Cone, A. L., Ndiokho, I. B., & Bruchas, M. R. (2021). Parabrachial opioidergic projections to preoptic hypothalamus mediate behavioral and physiological thermal defenses. *ELife, 10*. doi:10.7554/*eLife*.60779

Vong, L., Ye, C., Yang, Z., Choi, B., Chua, S., Jr., & Lowell, B. B. (2011). Leptin action on GABAergic neurons prevents obesity and reduces inhibitory tone to POMC neurons. *Neuron, 71*(1), 142-154. doi:10.1016/j.neuron.2011.05.028

Yang, W. Z., Du, X., Zhang, W., Gao, C., Xie, H., Xiao, Y.,… Fu, X. (2020). Parabrachial neuron types categorically encode thermoregulation variables during heat defense. *Science advances, 6*(36), eabb9414.

Zhang, B., Qiu, L., Xiao, W., Ni, H., Chen, L., Wang, F.,… Gao, Z. (2021). Reconstruction of the Hypothalamo-Neurohypophysial System and Functional Dissection of Magnocellular Oxytocin Neurons in the Brain. *Neuron, 109*(2), 331-346 e337. doi:10.1016/j.neuron.2020.10.032

[Editors’ note: further revisions were suggested prior to acceptance, as described below.]

The manuscript has been improved but there are some remaining issues that need to be addressed, as outlined below:Please address the important points noted by Reviewer #1 below. Maps of fiber placements and viral expression should be included in all relevant experiments.Please ensure your manuscript complies with the eLife policies for statistical reporting: https://reviewer.elifesciences.org/author-guide/full "Report exact p-values wherever possible alongside the summary statistics and 95% confidence intervals. These should be reported for all key questions and not only when the p-value is less than 0.05." in particular, statistics were not readily identified for Figure 1P, 3, N, O, Q, R.Reviewer #1 (Recommendations for the authors):The authors addressed most of my concerns.However, I still have a couple of minor concerns/recommendations:1. Figure 4 D-H – During the rat exposure behavioral test, the authors show that optogenetic inhibition of LPBCCK neurons results in increases in entries and time in the 'danger zone.' However, are there any changes in latencies to flight and flight speed after silencing these neurons?

We appreciate this question from the reviewer. We have analyzed the latencies to flight and flight speed from the ‘danger zone’ after silencing LPBCCK neurons. Our data showed that silencing LPBCCK neurons did not significantly change the latencies and flight speed, albeit there is a tendency for longer latencies.

**Author response image 4. sa2fig4:** Analysis of flight latency and speed after optic inhibition of LPBCCK neurons using GtACR1 virus. (For latency, p = 0.1866, t = 1.394, df = 13; unpaired *t* test; for speed, p = 0.6926, t = 0.4043, df = 13; unpaired *t* test).

2. For rigor purposes, the author should show anatomical maps displaying fiber placements for all the optogenetics and fiber photometry experiments.

We thank the reviewer for this suggestion, we have included all the figures showing the fiber placements in to Figure 2-6 in the main text of our manuscript. We have also uploaded these images into the source data.

3. Figure 3F is missing the y-axis.

We appreciate the reviewer for this important question and we have corrected the figure.

**Author response table 3. sa2table3:** Number of labeled cells.

Number of labeled cells
animal 1	animal 2	animal 3	total average
region	slice 1	slice 2	slice 3	total	percentage	average	region	slice 1	slice 2	slice 3	total	percentage	average	region	slice 1	slice 2	slice 3	total	percentage	average
CC	94	103	95	292	25.75%	97	CC	82	93	75	250	21.82%	83	CC	80	95	76	251	24.25%	84	88
DI	11	34	21	66	5.82%	22	DI	18	23	17	58	5.06%	19	DI	27	32	28	87	8.41%	29	23
LPB	71	103	69	243	21.43%	81	LPB	94	102	91	287	25.04%	96	LPB	67	82	63	212	20.48%	71	83
PAG	26	34	19	79	6.97%	26	PAG	11	23	15	49	4.28%	16	PAG	12	19	16	47	4.54%	16	19
MO	137	175	142	454	40.04%	151	MO	166	172	164	502	43.80%	167	MO	139	157	142	438	42.32%	146	155